# The Folded Spin-$1/2$ XXZ Model:
# I. Diagonalisation, Jamming, and Ground State Properties

L. Zadnik[1] and M. Fagotti[1]

**1** Université Paris-Saclay, CNRS, LPTMS, 91405, Orsay, France
*maurizio.fagotti@universite-paris-saclay.fr

September 22, 2020

## Abstract

**We study an effective Hamiltonian generating time evolution of states on intermediate time scales in the strong-coupling limit of the spin-$1/2$ XXZ model. To leading order, it describes an integrable model with local interactions. We solve it completely by means of a coordinate Bethe Ansatz that manifestly breaks the translational symmetry. We demonstrate the existence of exponentially many jammed states and estimate their stability under the leading correction to the effective Hamiltonian. Some ground state properties of the model are discussed.**

## 1 Introduction

Thermalisation in isolated quantum systems has received enormous attention in the last few decades (see review articles [1–3] and references therein), partly due to the advancement of the experimental techniques that allow one to manipulate a large number of quantum particles [4, 5]. A particularly important question in this regard has been of how the conserved quantities affect the dynamics of a system. In integrable models this was clarified only recently with the introduction of so-called Generalised Gibbs Ensembles (GGE) [6–8], which incorporate the constraints of infinitely many conserved charges [9, 10] and describe the expectation values of the local observables at asymptotically large times [11]. In the attempt to apply a similar theory in the presence of inhomogeneities, a generalised hydrodynamic description (GHD) was developed [12, 13]. To leading order, which is the one better understood so far, the theory describes the time evolution of a class of inhomogeneous states by means of a "classical" continuity equation[1], which, together with Bethe Ansatz, predicts the expectation values of the local observables on Euler scales. On the other hand, the dynamics of isolated integrable systems on intermediate time scales remains an intriguing open problem. Strong coupling limits in this respect play an important role. In particular, unusually slow symmetry restoration typically indicates that the system can be seen as a perturbation of an effective one, in which the symmetry is never restored. Knowledge of the effective system and its symmetries is thus necessary to understand pre-relaxation behaviour [15].

Strong coupling limits of quantum models have been thoroughly investigated at equilibrium, especially after the seminal work by MacDonald, Girvin, and Yoshioka [16], which proposed an asymptotic perturbation theory for a class of systems that includes the Hubbard model. Later, an alternative approach to calculate the thermal correlation functions in strong coupling limits was developed; it is based on investigating the spectrum of constrained Hamiltonians describing particles with a finite radius and acting on the reduced Hilbert space [17–20]. Strong coupling descriptions of time evolution are, however, still under investigation. Based on the results of MacDonald *et. al.*, we consider an asymptotic formulation of quantum mechanics that attaches the fast oscillatory part of the dynamics to the operators, letting the state evolve gently in time under an effective Hamiltonian. We term this formulation "asymptotic folded picture", to emphasise that the spectrum of the effective "folded Hamiltonian" is the same as that of the original Hamiltonian, modulo a typical energy

---

[1]The theory is classical in the sense that $\hbar$ does not appear in the equation explicitly.

proportional to the largest coupling constant. In the folded picture localised operators are asymptotically mapped to quasilocalised ones, hence the effective Hamiltonian captures the essential features of the original dynamics.

As an example, we consider the large anisotropy limit of the spin-1/2 XXZ chain and present a Bethe Ansatz solution of the model described by the corresponding local folded Hamiltonian. Since the XXZ model is integrable for any value of the anisotropy parameter, one could address this problem by applying the Schrieffer-Wolff transformation of Ref. [16] to the eigenstates of the original model – see also Ref. [21]. This approach, however, conceals the extra symmetries that generally emerge in strong coupling limits. For example, at large anisotropy the time evolution generated by the effective Hamiltonian of the XXZ model does not restore one-site shift invariance if that is broken in the initial state [15]. Being translationally invariant, the standard Bethe Ansatz basis is then arguably inappropriate for investigations into intermediate time scales. We propose to overcome this problem by a change of basis that explicitly breaks one-site shift translational symmetry. We stress that the choice of a basis is important because the theories describing relaxation are usually presented in a way that is, in fact, basis dependent; hidden symmetries can spoil the predictions, despite the theories being correct – see, *e.g.*, Ref. [15]. The dissimilarity between our solution and the standard one is manifested in the striking fact that the total spin along the anisotropy axis, which counts the number of rapidities in the standard Bethe Ansatz solution, is not even diagonal in the basis that we consider.

We will show that the folded XXZ Hamiltonian exhibits a richer structure than the XXZ Hamiltonian, reflecting the non-abelian integrability of the folded model. In particular, the Hamiltonian can be diagonalised by introducing two species of impenetrable particles that interact via a diagonal scattering matrix. This implies the conservation of the particle configuration, which turns out to have visible effects. The Hilbert space divides into three exponentially large sectors: (a), a fully interacting one, with a mixed population of both types of particles, (b), two noninteracting sectors, where only a single particle species is present, and, (c), a sector of jammed inhomogeneous eigenstates in which the particles are densely packed and can not move. Remarkably, a subset of the latter states is stable under the perturbation by the leading correction to the strong coupling limit.

## Summary

**Section 2.** We work out an iterative solution of the strong coupling expansion firstly proposed in Ref. [16] and define the "folded picture" and the "folded Hamiltonian". We also exhibit the first orders of the formal asymptotic expansion of the folded Hamiltonian and the corresponding unitary transformation.

**Section 3.** We focus on the folded Hamiltonian of the XXZ model. We identify a duality transformation that compresses the range of the operator in the limit of infinite anisotropy.

**Section 4.** We define two species of particles that sit on macrosites consisting of two neighbouring sites and show that their configuration is a topological invariant. We exhibit an exponentially large number of jammed product states with zero energy and assess their stability under the leading strong-coupling correction.

**Section 5.** The (dual) folded XXZ Hamiltonian is diagonalised via a coordinate Bethe Ansatz

based on the two species of particles.

**Section 6.** We identify the ground state of the folded XXZ Hamiltonian and compute its energy. At a given energy density we also compute the maximal number of consecutive particles of the same species, *i.e.*, the size of a domain.

# 2 Dynamics in strong coupling limits: the folded picture

A large class of quantum lattice models are described by Hamiltonians $\mathbf{H}(\kappa)$ that can be decomposed as [16, 23–30]

$$\mathbf{H}(\kappa) = \mathbf{H}_F + \sum_{m=1}^{q} \left( \mathbf{F}_m + \mathbf{F}_m^\dagger \right) + \kappa^{-1}\mathbf{H}_I, \tag{1}$$

where $q$ is a finite integer, $\kappa$ the inverse coupling constant (a perturbative parameter in the strong coupling limit), while operators $\mathbf{H}_F$, $\mathbf{H}_I$, and $\mathbf{F}_m$, for $m = 1, 2\ldots, q$, satisfy

$$[\mathbf{H}_I, \mathbf{H}_F] = 0, \qquad [\mathbf{H}_I, \mathbf{F}_m] = mJ\mathbf{F}_m. \tag{2}$$

Constant $J$ has the units of energy and typically corresponds to an overall factor in the Hamiltonian $\mathbf{H}(\kappa)$. Algebraic relations (2) imply that, if $|\phi\rangle$ and $|\phi'\rangle$ are simultaneous eigenstates of $\mathbf{H}_I$ and $\mathbf{H}_F$, either $\langle\phi|\mathbf{F}_m|\phi'\rangle = 0$ or $\langle\phi|\mathbf{H}_I|\phi\rangle - \langle\phi'|\mathbf{H}_I|\phi'\rangle = mJ$. Thus, the eigenstates of $\mathbf{H}(\kappa)$ can be chosen to belong to sectors where the spectrum of $\mathbf{H}_I$ is equally spaced, the eigenvalue differences being multiples of $J$. This is the basic reason why, in all the relevant examples, $\mathbf{H}_I$ turns out to be a very simple operator, for example, the projection of the total spin in a given direction. For the sake of simplicity, we will assume that there is a single sector, *i.e.*, $\mathbf{H}_I$ has equally spaced eigenvalues and can be interpreted as a "number operator".

The class of models that allow for decomposition (1) includes both integrable and generic systems. Examples with $q = 1$ are the Hubbard model [16], the Heisenberg XYZ model, the Ising model in transverse and longitudinal magnetic field, the Kitaev model [24], the ANNNI model, and the Bloch-Stark MBL Hamiltonian [25, 26]. While the focus of this paper is on the Heisenberg XXZ model, in which $q = 1$, this section aims to present the framework for a generic $q$.

To the best of our knowledge, Ref. [16] was the first to observe that, in the strong coupling limit $\kappa \ll 1$ with $q = 1$, algebra (2) can be exploited within a perturbation theory to construct a weak coupling model with the spectrum of the original Hamiltonian. In this section we propose a formulation of quantum mechanics that exploits the results of Ref. [16] to investigate time evolution on an intermediate time scale. Specifically, we define the *asymptotic folded picture* as the formulation in which operators $\mathbf{O}$ and state $|\Psi(t)\rangle$ time evolve according to

$$\mathbf{O}_F(t) := e^{i\kappa\mathbf{B}_{z_t}(\kappa)}e^{i\kappa^{-1}\mathbf{H}_I t}\mathbf{O}e^{-i\kappa^{-1}\mathbf{H}_I t}e^{-i\kappa\mathbf{B}_{z_t}(\kappa)}, \quad |\Psi(t)\rangle_F := e^{-i\mathbf{H}_F(\kappa)t}e^{i\kappa\mathbf{B}_1(\kappa)}|\Psi(0)\rangle, \tag{3}$$

where $z_t = e^{iJt/\kappa}$ and the following conditions hold:

1. the "folded Hamiltonian" $\mathbf{H}_F(\kappa)$ is a functional of $\{\mathbf{F}_m\}_{m=1}^{q}$, $\{\mathbf{F}_m^\dagger\}_{m=1}^{q}$ and $\mathbf{H}_F \equiv \mathbf{H}_F(0)$ such that $[\mathbf{H}_F(\kappa), \mathbf{H}_I] = 0$,

2. $\mathbf{B}_z(\kappa)$ is a functional of the form $\mathbf{B}[\{z^m\mathbf{F}_m\}_{m=1}^{q}, \{z^{-m}\mathbf{F}_m^\dagger\}_{m=1}^{q}, \mathbf{H}_F; \kappa]$,

3. $\mathbf{B}_z(\kappa)$ is (assumed to be) holomorphic in $z$, in an open annulus enclosing the complex unit circle $S^1 = \{z \in \mathbb{C}; |z| = 1\}$, and its Laurent series is

$$\mathbf{B}_z(\kappa) = \sum_{n \in \mathbb{Z} \setminus \{0\}} \mathbf{A}_n(\kappa) z^n, \qquad \text{with} \qquad \mathbf{A}_{-n}(\kappa) = \mathbf{A}_n^\dagger(\kappa). \tag{4}$$

Within this picture, the part of the dynamics that oscillates rapidly with a period $2\pi\kappa/J$ is applied to the operators, whereas the states time evolve under a weak-coupling Hamiltonian that conserves the strong coupling term $\mathbf{H}_I$. Since $\mathbf{H}_I$ is generally trivial, this allows for a non-perturbative definition of a reference state (for example, the ground state of $\mathbf{H}_I$), over which we can define and organise elementary excitations.

Using algebra (2) and the conditions satisfied by $\mathbf{B}_z(\kappa)$, we see that formulation (3) of the asymptotic folded picture is equivalent to expressing the time evolution operator as

$$e^{-i\mathbf{H}(\kappa)t} = e^{-i\kappa\mathbf{B}_1(\kappa)} e^{-i[\mathbf{H}_F(\kappa) + \kappa^{-1}\mathbf{H}_I]t} e^{i\kappa\mathbf{B}_1(\kappa)}. \tag{5}$$

Hence, we can identify $\mathbf{H}_F(\kappa) + \kappa^{-1}\mathbf{H}_I$ with the effective Hamiltonian equivalent to $\mathbf{H}(\kappa)$, and $e^{i\kappa\mathbf{B}_1(\kappa)}$ with the corresponding unitary transformation, both considered in Ref. [16]. As an interesting fact we observe that the unitary transformation (5) is equivalent to the end point $\mathbf{\Phi}_1 = \mathbf{H}_F(\kappa) + \kappa^{-1}\mathbf{H}_I$ of the one-parameter flow $\mathbf{\Phi}_s$, defined for $s \in [0, 1]$ as

$$\mathbf{\Phi}_s = e^{i\kappa\mathbf{B}_s(\kappa)} \mathbf{H}(\kappa) e^{-i\kappa\mathbf{B}_s(\kappa)}, \tag{6}$$

and subject to the initial condition $\mathbf{\Phi}_0 = \mathbf{H}(\kappa)$. Roughly speaking, operator $\mathbf{B}_s(\kappa)$ acts as a generator of the flow equations

$$\partial_s \mathbf{\Phi}_s = [\boldsymbol{\eta}_s, \mathbf{\Phi}_s], \qquad \boldsymbol{\eta}_s = i\kappa \int_0^1 d\lambda\, e^{i\lambda\kappa\mathbf{B}_s(\kappa)} [\partial_s \mathbf{B}_s(\kappa)] e^{-i\lambda\kappa\mathbf{B}_s(\kappa)}, \tag{7}$$

which have been introduced by Wegner [22] as a way to suppress the off-diagonal terms of a Hamiltonian by means of a continuous unitary transformation. In our case, however, the ending point $\mathbf{H}_F(\kappa) + \kappa^{-1}\mathbf{H}_I$ of the flow is far from being a diagonal operator. In fact, it describes a generic interacting model.

Both the folded Hamiltonian $\mathbf{H}_F(\kappa)$ and the unitary transformation $e^{i\kappa\mathbf{B}_1(\kappa)}$ can be worked out asymptotically. In Appendix A we report the recurrence relation for the coefficients of the asymptotic expansion

$$\mathbf{A}_\ell(\kappa) = \sum_{j=0}^\infty \mathbf{A}_{\ell,j} \kappa^j, \tag{8}$$

as well as the lowest orders of the expansion for $q = 2$ (for $q = 1$ see also Ref. [27]). Here we will focus on the zeroth order and the leading correction, which yield

$$\mathbf{H}_F(\kappa) = \mathbf{H}_F + \kappa \mathbf{H}_F' + \mathcal{O}(\kappa^2) = \mathbf{H}_F + \frac{\kappa}{J}\left([\mathbf{F}_1, \mathbf{F}_1^\dagger] + \frac{1}{2}[\mathbf{F}_2, \mathbf{F}_2^\dagger]\right) + \mathcal{O}(\kappa^2) \tag{9}$$

for the folded Hamiltonian (the $\kappa^2$ correction is reported in Eq. (156) of Appendix A), and

$$e^{i\kappa\mathbf{B}_1(\kappa)} = e^{\frac{\kappa}{J}[\mathbf{F}_1 + \mathbf{F}_2 - \mathbf{F}_1^\dagger - \mathbf{F}_2^\dagger] + \mathcal{O}(\kappa^2)} \tag{10}$$

for the unitary transformation.

Finally, we note that $[\mathbf{H}_I, \mathbf{H}_F(\kappa)] = 0$ and Eq. (5) imply that $e^{-i\kappa\mathbf{B}_1(\kappa)} \mathbf{H}_F(\kappa) e^{i\kappa\mathbf{B}_1(\kappa)}$ is a conservation law for $\mathbf{H}(\kappa)$; by checking the first orders of its asymptotic expansion using Eqs. (9) and (10), we realise that it is exactly the charge perturbatively constructed in Ref. [31].

## 2.1 Integrability

One of the interesting aspects of the asymptotic expansion (9) is that the leading order $\mathbf{H}_F$ of the folded Hamiltonian $\mathbf{H}_F(\kappa)$ can be integrable even if the full $\mathbf{H}(\kappa)$ pertains to a generic system. Moreover, since $\mathbf{H}_F(\kappa)$ commutes with $\mathbf{H}_I$, which is generally a trivial operator, a simple reference state can always be defined and the asymptotic model is likely to be solvable by coordinate Bethe Ansatz. Remarkably, the leading order of the folded Hamiltonian has usually more symmetries than the original model, starting from the $U(1)$ symmetry that is associated with $\mathbf{H}_I$ by construction. In addition, every partial symmetry acting like $\mathbf{U}\mathbf{H}(\kappa)\mathbf{U}^\dagger = \mathbf{H}(-\kappa)$ becomes an exact symmetry for $\mathbf{H}_F$.

Especially interesting phenomenology arises when the folded Hamiltonian is asymptotically non-abelian integrable, that is, when it possesses a non-abelian set of quasilocal conservation laws. The practical effect of non-abelian integrability is that states sharing a complete set of integrals of motion remain locally distinguishable even at late times [15]. This is manifested in a physically relevant degeneracy of stationary states even at extensively high energies, in turn implying that different integrable models share the leading order of the folded Hamiltonian. In this situation, constructing the folded Hamiltonian as a limit of an integrable system is likely to hide the emergent symmetries. It is arguably more effective to consider the folded Hamiltonian as a different model, redefining, if advantageous, reference state and excitations. This is the point of view that we embrace in the next part of the paper, where we work out a Bethe Ansatz solution to the asymptotics of the folded XXZ Hamiltonian.

On the other hand, even if the original model is integrable for any value of $\kappa$, it is not reasonable to expect the folded Hamiltonian truncated at a given order to remain integrable. However, if the truncation still has more symmetries than the original Hamiltonian, we envisage the possibility to find distinct integrable systems sharing the first orders of the expansion. In that case, the dynamics of states in the strong coupling regime can exhibit a cascade of pre-relaxation plateaux.

## 2.2 Examples

**Generic model with integrable $\mathbf{H}_F(0)$.** An example of a generic system with an asymptotically integrable folded Hamiltonian is the XYZ spin-1/2 chain in an external magnetic field. It is described by the Hamiltonian

$$\mathbf{H} = J\sum_\ell (1+\gamma)\boldsymbol{\sigma}_\ell^x \boldsymbol{\sigma}_{\ell+1}^x + (1-\gamma)\boldsymbol{\sigma}_\ell^y \boldsymbol{\sigma}_{\ell+1}^y + \Delta\boldsymbol{\sigma}_\ell^z \boldsymbol{\sigma}_{\ell+1}^z - J\frac{h}{4}\sum_\ell \boldsymbol{\sigma}_\ell^z. \tag{11}$$

In the strong coupling limit $h \to \infty$, the folded picture is characterised by the XXZ Hamiltonian

$$\mathbf{H}_F \equiv \mathbf{H}_F(0) = J\sum_\ell \boldsymbol{\sigma}_\ell^x \boldsymbol{\sigma}_{\ell+1}^x + \boldsymbol{\sigma}_\ell^y \boldsymbol{\sigma}_{\ell+1}^y + \Delta\boldsymbol{\sigma}_\ell^z \boldsymbol{\sigma}_{\ell+1}^z, \tag{12}$$

which is integrable. Identifying $h$ with $\kappa^{-1}$ we find $q = 1$ and

$$\mathbf{F}_1 = 2J\gamma \sum_\ell \boldsymbol{\sigma}_\ell^- \boldsymbol{\sigma}_{\ell+1}^-, \tag{13}$$

so that the leading correction reads

$$\frac{\kappa}{J}[\mathbf{F}_1, \mathbf{F}_1^\dagger] = -\frac{2J\gamma^2}{h}\sum_\ell \boldsymbol{\sigma}_{\ell-1}^x \boldsymbol{\sigma}_\ell^z \boldsymbol{\sigma}_{\ell+1}^x + \boldsymbol{\sigma}_{\ell-1}^y \boldsymbol{\sigma}_\ell^z \boldsymbol{\sigma}_{\ell+1}^y + \boldsymbol{\sigma}_\ell^z. \tag{14}$$

To the best of our knowledge this perturbation breaks integrability, however, we have not investigated whether integrability can be restored by the addition of the subleading terms.

**Generic model with non-abelian integrable $\mathbf{H}_F(0)$.** An example of a generic model with a non-abelian integrable strong coupling limit is given by the quantum Ising model in longitudinal and transverse fields, described by the Hamiltonian

$$\mathbf{H} = -\frac{J}{2}\sum_\ell \boldsymbol{\sigma}_\ell^x\boldsymbol{\sigma}_{\ell+1}^x + g\boldsymbol{\sigma}_\ell^x + h\boldsymbol{\sigma}_\ell^z. \tag{15}$$

In the limit $h \to \infty$ we identify $\kappa = h^{-1}$, $q = 2$, and

$$\mathbf{H}_F = -\frac{J}{4}\sum_\ell \boldsymbol{\sigma}_\ell^x\boldsymbol{\sigma}_{\ell+1}^x + \boldsymbol{\sigma}_\ell^y\boldsymbol{\sigma}_{\ell+1}^y, \qquad \mathbf{F}_1 = -\frac{Jg}{2}\sum_\ell \boldsymbol{\sigma}_\ell^-, \qquad \mathbf{F}_2 = -\frac{J}{2}\sum_\ell \boldsymbol{\sigma}_\ell^-\boldsymbol{\sigma}_{\ell+1}^-. \tag{16}$$

The expansion in $h^{-1}$ of the folded Hamiltonian reads

$$\mathbf{H}_F(h^{-1}) = \frac{J}{4}\Big[\frac{1+8g^2}{8h^2}-1\Big]\sum_\ell \mathbf{K}_{\ell,\ell+1} - \frac{J}{16h}\sum_\ell(\mathbf{K}_{\ell-1,\ell+1}+2[1+2g^2])\boldsymbol{\sigma}_\ell^z - \frac{J}{32h^2}\sum_\ell \mathbf{K}_{\ell-1,\ell+2}\boldsymbol{\sigma}_\ell^z\boldsymbol{\sigma}_{\ell+1}^z$$

$$- \frac{Jg^2}{2h^2}\sum_\ell \boldsymbol{\sigma}_\ell^z\boldsymbol{\sigma}_{\ell+1}^z + \mathcal{O}(h^{-3}), \quad (17)$$

where $\mathbf{K}_{n,m} = \boldsymbol{\sigma}_n^x\boldsymbol{\sigma}_m^x + \boldsymbol{\sigma}_n^y\boldsymbol{\sigma}_m^y$. At the leading order $\mathbf{H}_F$ describes the XX model (first term), which is noninteracting and known to be non-abelian integrable [15]. Quite interestingly, the folded Hamiltonian remains noninteracting even at $\mathcal{O}(h^{-1})$ (second term). The $\mathcal{O}(h^{-1})$ correction, however, breaks the non-abelian integrability. Interactions appear only at $\mathcal{O}(h^{-2})$ (second line).

**Integrable model with quasilocal $\mathbf{H}_F(\kappa)$.** Strong coupling limits in noninteracting models generally result in quasilocal folded Hamiltonians. We mention, for instance, the quantum Ising model in a transverse field

$$\mathbf{H} = -\frac{J}{4}\sum_\ell \boldsymbol{\sigma}_\ell^x\boldsymbol{\sigma}_{\ell+1}^x + h\boldsymbol{\sigma}_\ell^z, \tag{18}$$

which corresponds to setting $g = 0$ in the previous example. Since this Hamiltonian can be represented by a quadratic operator of spinless fermions, it is easy to find a unitary transformation mapping $\mathbf{H}$ into a $U(1)$ symmetric operator that commutes with $\sum_\ell \boldsymbol{\sigma}_\ell^z$ – see Appendix B. Specifically, we have

$$\mathbf{H} = e^{-ih^{-1}\mathbf{B}_1(h^{-1})}\Big[\mathbf{H}_F(h^{-1}) - \frac{Jh}{4}\sum_\ell \boldsymbol{\sigma}_\ell^z\Big]e^{ih^{-1}\mathbf{B}_1(h^{-1})}, \tag{19}$$

with

$$\mathbf{B}_1(h^{-1}) = -\frac{i}{16}\sum_{\ell,m}\int_{-\pi}^{\pi}\frac{\mathrm{d}k}{2\pi}e^{i(\ell-n)k}\log\frac{1-h^{-1}e^{ik}}{1-h^{-1}e^{-ik}}(\mathbf{a}_{2\ell-1}\mathbf{a}_{2m-1}-\mathbf{a}_{2\ell}\mathbf{a}_{2m}),$$

$$\mathbf{H}_F(h^{-1}) = \frac{J}{4}\sum_{\ell,n}\int_{-\pi}^{\pi}\frac{\mathrm{d}k}{2\pi}e^{i(\ell-m)p}h\frac{1-\sqrt{1-2h^{-1}\cos k+h^{-2}}}{2}(\mathbf{a}_{2\ell}\mathbf{a}_{2m-1}-\mathbf{a}_{2\ell-1}\mathbf{a}_{2m}),$$

$$(20)$$

and

$$\mathbf{a}_{2\ell-1} := \prod_{j<\ell} \sigma_j^z \sigma_\ell^x, \qquad \mathbf{a}_{2\ell} := \prod_{j<\ell} \sigma_j^z \sigma_\ell^y, \qquad \{\mathbf{a}_\ell, \mathbf{a}_n\} = 2\delta_{\ell n}\mathbf{1}. \qquad (21)$$

A posteriori we see that the expansion in $h^{-1}$ converges for $h^{-1} < h_c = 1$, where $h_c$ is the magnetic field corresponding to the Ising phase transition. In particular, $\mathbf{H}_F(h^{-1})$ and $\mathbf{B}_1(h^{-1})$ are quasilocal with typical range $(\log h)^{-1}$.

**Interacting $\mathbf{H}_F(0)$ that breaks one-site shift invariance.** In this class we identify the XYZ spin-1/2 chain described by the Hamiltonian (11) with $h = 0$. In the strong coupling limit $\Delta \to \infty$, the folded picture is characterised by the Hamiltonian

$$\mathbf{H}_F = J \sum_\ell \frac{1+\gamma}{2}(\sigma_\ell^x \sigma_{\ell+1}^x + \sigma_{\ell-1}^z \sigma_\ell^y \sigma_{\ell+1}^y \sigma_{\ell+2}^z) + \frac{1-\gamma}{2}(\sigma_\ell^y \sigma_{\ell+1}^y + \sigma_{\ell-1}^z \sigma_\ell^x \sigma_{\ell+1}^x \sigma_{\ell+2}^z). \qquad (22)$$

The reader can easily check that $\mathbf{H}_F$ commutes with

$$\mathbf{H}_I^- = \frac{1}{4} \sum_\ell (-1)^\ell \sigma_\ell^z \sigma_{\ell+1}^z, \qquad (23)$$

which breaks one-site shift invariance and causes the system to retain the memory of one-site shift asymmetry. In the rest of the paper we will focus on the special case $\gamma = 0$, which exhibits even more symmetries that give rise to exceptional degeneracies and jamming. We leave the study of Hamiltonian (22) with $\gamma \neq 0$ to future investigations.

## 3 The folded XXZ Hamiltonian in the strong coupling limit

The folded Hamiltonians listed in the previous section are all asymptotically integrable, and, except for the last one, have leading terms $\mathbf{H}_F$ that describe well-known models. Here, we focus on the folded Hamiltonian of the XXZ model at large anisotropy, which turns out to be a meagrely studied integrable system with striking features.

The XXZ spin-1/2 chain is described by the Hamiltonian

$$\mathbf{H} = J \sum_{\ell=1}^L \sigma_\ell^x \sigma_{\ell+1}^x + \sigma_\ell^y \sigma_{\ell+1}^y + \Delta \sigma_\ell^z \sigma_{\ell+1}^z, \qquad (24)$$

where periodic boundary conditions $\sigma_{L+1}^\alpha = \sigma_1^\alpha$ are understood. It is arguably the simplest model capturing the antiferromagnetic properties of crystals with effective one-dimensional magnetic behaviour, like the $KCuF_3$ compound [32]. From a theoretical point of view, it is integrable for any value of the anisotropy $\Delta$. The ground state phase diagram exhibits antiferromagnetism for $\Delta > 1$, ferromagnetism for $\Delta < 1$, and a critical phase for $\Delta \in [-1, 1]$. We are interested in the limit of large anisotropy. In the folded picture we identify $\kappa$ with $(4\Delta)^{-1}$, and the operators $\mathbf{H}_I$, $\mathbf{F}_1 \equiv \mathbf{F}$, and $\mathbf{H}_F$ are given by

$$\mathbf{H}_I = \frac{J}{4} \sum_{\ell=1}^L \sigma_\ell^z \sigma_{\ell+1}^z, \qquad (25)$$

$$\mathbf{F} = \frac{J}{2} \sum_{\ell=1}^L (\sigma_\ell^x \sigma_{\ell+1}^x + \sigma_\ell^y \sigma_{\ell+1}^y) \frac{1 - \sigma_{\ell-1}^z \sigma_{\ell+2}^z}{2} + i(\sigma_\ell^x \sigma_{\ell+1}^y - \sigma_\ell^y \sigma_{\ell+1}^x) \frac{\sigma_{\ell+2}^z - \sigma_{\ell-1}^z}{2}, \qquad (26)$$

$$\mathbf{H}_F = J \sum_{\ell=1}^{L} \frac{1 + \sigma_{\ell-1}^z \sigma_{\ell+2}^z}{2} (\sigma_\ell^x \sigma_{\ell+1}^x + \sigma_\ell^y \sigma_{\ell+1}^y). \tag{27}$$

To the best of our knowledge, until now the folded Hamiltonian has been considered only in the combination $\mathbf{H}_F + 4\Delta\mathbf{H}_I$ in the limit of large $\Delta$. That is, indeed, the Hamiltonian unitarily equivalent to the XXZ model in the limit of large anisotropy. Since $\mathbf{H}_I$ commutes with $\mathbf{H}_F$, the two operators are diagonal in the same basis. As discussed in Ref. [21], $\mathbf{H}_F$ could thus be diagonalised by transforming the eigenstates of the XXZ Hamiltonian and expanding the corresponding eigenvalues in the limit of large $\Delta$. The drawback to this approach is that the solution is obtained as a limit of a model with less symmetries, forcing one to deal with a redundantly complicated structure (analogously to solving the quantum harmonic oscillator as a limit of the anharmonic one). In order to circumvent these complications, we lift the folded XXZ Hamiltonian into an independent model, providing an ab initio solution that will allow us to unveil some of the properties that make it essentially different from the XXZ model.

## 3.1 Symmetries and short-range conservation laws

The folded Hamiltonian inherits the following symmetries of the XXZ model: one-site shift invariance, reflection symmetry, spin flip symmetry ($[\prod_{\ell=1}^{L} \sigma_\ell^\alpha, \mathbf{H}] = 0$, for $\alpha = x, y, z$) and conservation of the total magnetisation along the $z$-axis

$$\mathbf{S}^z = \frac{1}{2} \sum_{\ell=1}^{L} \sigma_\ell^z. \tag{28}$$

By construction, it also commutes with $\mathbf{H}_I$ (25). The first unusual symmetry of $\mathbf{H}_F$ is associated with the conservation of the staggered version of $\mathbf{H}_I$,

$$\mathbf{H}_I^- = \frac{J}{4} \sum_{\ell=1}^{L} (-1)^\ell \sigma_\ell^z \sigma_{\ell+1}^z, \tag{29}$$

which breaks one-site shift invariance. This symmetry is, in fact, broken at a higher level. In particular, Ref. [15] showed that the model has noninteracting sectors (characterised by $\sigma_{2\ell-1}^z \sigma_{2\ell}^z = -1$) that break translational invariance and can be effectively described by a noninteracting Hamiltonian with a non-abelian set of local conservation laws. The origin of these sectors will be clarified in Section 4.1.

The model described by the Hamiltonian (27) has other short-range shift-invariant conservation laws. One can identify them by imposing the vanishing of the commutator between a generic local translationally invariant operator and $\mathbf{H}_F$. We denote them by

$$\mathbf{Q}_n^\pm = \sum_{\ell=1}^{L} \mathbf{q}_{n,\ell}^\pm, \tag{30}$$

where $\mathbf{q}_{n,\ell}^\pm$ is the local density acting on sites $\ell, \dots, \ell + 2n + \frac{1 \pm 1}{2}$, $n \in \mathbb{Z}_> \equiv \mathbb{N}$, while the sign specifies whether the charge is even (+) or odd (−) under reflection symmetry. The densities

of the first few conservation laws read

$$\mathbf{q}_{1,\ell}^{-} = -\frac{J}{2}\left(\mathbf{D}_{\ell,\ell+1}\boldsymbol{\sigma}_{\ell+2}^{z} + \boldsymbol{\sigma}_{\ell}^{z}\mathbf{D}_{\ell+1,\ell+2}\right),$$

$$\mathbf{q}_{1,\ell}^{+} = \frac{J}{2}\left(\mathbf{1} + \boldsymbol{\sigma}_{\ell}^{z}\boldsymbol{\sigma}_{\ell+3}^{z}\right)\mathbf{K}_{\ell+1,\ell+2},$$

$$\mathbf{q}_{2,\ell}^{-} = \frac{J}{4}\Big(\mathbf{D}_{\ell,\ell+2}[\boldsymbol{\sigma}_{\ell+1}^{z} + \boldsymbol{\sigma}_{\ell+3}^{z}] + \mathbf{D}_{\ell+1,\ell+3}[\boldsymbol{\sigma}_{\ell}^{z} + \boldsymbol{\sigma}_{\ell}^{z}\boldsymbol{\sigma}_{\ell+2}^{z}\boldsymbol{\sigma}_{\ell+4}^{z}] - \mathbf{D}_{\ell,\ell+1}\mathbf{K}_{\ell+2,\ell+3}\boldsymbol{\sigma}_{\ell+4}^{z} \tag{31}$$
$$- \boldsymbol{\sigma}_{\ell}^{z}\mathbf{K}_{\ell+1,\ell+2}\mathbf{D}_{\ell+3,\ell+4}\Big),$$

$$\mathbf{q}_{2,\ell}^{+} = -\frac{J}{4}\Big(\mathbf{K}_{\ell,\ell+2}[\mathbf{1} + \boldsymbol{\sigma}_{\ell+1}^{z}\boldsymbol{\sigma}_{\ell+3}^{z}] + \mathbf{D}_{\ell,\ell+1}\mathbf{D}_{\ell+2,\ell+3} + [\boldsymbol{\sigma}_{\ell}^{z}\boldsymbol{\sigma}_{\ell+2}^{z} + \boldsymbol{\sigma}_{\ell}^{z}\boldsymbol{\sigma}_{\ell+4}^{z}]\mathbf{K}_{\ell+1,\ell+3}$$
$$- \boldsymbol{\sigma}_{\ell}^{z}\mathbf{K}_{\ell+1,\ell+2}\mathbf{K}_{\ell+3,\ell+4}\boldsymbol{\sigma}_{\ell+5}^{z}\Big),$$

where $\mathbf{K}_{n,m}$ is defined below Eq. (17) and $\mathbf{D}_{n,m} = \boldsymbol{\sigma}_{n}^{x}\boldsymbol{\sigma}_{m}^{y} - \boldsymbol{\sigma}_{n}^{y}\boldsymbol{\sigma}_{m}^{x}$ is a *Dzyaloshinskii-Moriya* term. Note that $\mathbf{Q}_{1}^{+} = \mathbf{H}_{F}$ is exactly the folded Hamiltonian.

Interestingly, the folded Hamiltonian $\mathbf{H}_{F}$ and other computed short-range charges $\mathbf{Q}_{n}^{\pm}$ commute with the transfer matrix of the four-vertex model, obtained from the XXZ model's transfer matrix in the $\Delta \to \infty$ (Ising) limit [18, 19]. The four-vertex model transfer matrix enables the algebraic Bethe Ansatz technique, which produces a basis of eigenstates corresponding to the nonzero energies of the following two equivalent projected Hamiltonians:

$$\mathbf{P}\,\mathbf{H}_{F}\,\mathbf{P} = \mathbf{P}\,\mathbf{H}_{\mathrm{XX}}\,\mathbf{P}.^{2} \tag{32}$$

Here, $\mathbf{H}_{\mathrm{XX}} = J\sum_{\ell=1}^{L}\mathbf{K}_{\ell,\ell+1}$ is the Hamiltonian of the XX model, while the projector

$$\mathbf{P} = \prod_{\ell=1}^{L}\left(\mathbf{1} - \frac{1 - \boldsymbol{\sigma}_{\ell}^{z}}{2}\frac{1 - \boldsymbol{\sigma}_{\ell+1}^{z}}{2}\right) \tag{33}$$

prevents neighbouring sites to be occupied simultaneously, thus inducing a hard-core repulsion [18–20]. Because of the latter, such projected Hamiltonians can only reproduce the low-energy sector of $\mathbf{H}_{F}$.[3] The eigenstates that are destroyed by the projector $\mathbf{P}$ unfortunately turn out to be inaccessible by the algebraic Bethe Ansatz that diagonalises the four-vertex transfer matrix.

Nevertheless, our numerical analysis is consistent with the existence of an infinite sequence of local charges, resembling the family of local conservation laws of the XXZ Hamiltonian. The latter set is known not to characterise the excited states completely: the remaining conservation laws are quasilocal and are, within the framework of algebraic Bethe Ansatz, generated by the transfer matrices associated with higher-spin representations of the Lax operator [33]. We will come back to the completeness of the local charges of the folded XXZ Hamiltonian after having worked out a coordinate Bethe Ansatz that produces a complete basis of eigenstates.

## 3.2  Duality transformation

In this section we exhibit and work out a duality transformation that, up to boundary terms, maps $\mathbf{H}_{I}$ into $\frac{J}{2}\mathbf{S}^{z}$ and reduces the range of the local density of $\mathbf{H}_{F}$ from four to three

---

[2] We note that these two Hamiltonians also commute with $\mathbf{H}_{F}$ and with the other charges $\mathbf{Q}_{n}^{\pm}$.

[3] That the reproduced sector contains the low-energy eigenstates of $\mathbf{H}_{F}$ is supported by the fact that the energy density and the Fermi velocity in the ground state of $\mathbf{H}_{F}$, computed in Section 6, coincide with those computed in Ref. [20] for the projected Hamiltonian $\mathbf{P}\,\mathbf{H}_{F}\,\mathbf{P}$.

neighbouring sites. The transformation consists of two steps[4]:

1. rotation around the $y$-axis

$$\mathbf{O} \mapsto e^{i\frac{\pi}{4}\sum_{\ell=1}^{L}\boldsymbol{\sigma}_\ell^y}\mathbf{O}e^{-i\frac{\pi}{4}\sum_{\ell=1}^{L}\boldsymbol{\sigma}_\ell^y}, \tag{34}$$

2. left antiperiodic pseudo-site shift of the Jordan-Wigner Majorana fermions (21)

$$\mathbf{a}_\ell \mapsto \begin{cases} -\mathbf{a}_{2L}, & \ell = 1, \\ \mathbf{a}_{\ell-1}, & \ell > 1. \end{cases} \tag{35}$$

We will call $\tilde{\mathbf{O}}$ the image of operator $\mathbf{O}$ under this transformation. In particular, it maps the folded Hamiltonian with periodic boundary conditions ($\boldsymbol{\sigma}_{L+n}^\alpha = \boldsymbol{\sigma}_n^\alpha$, for $\alpha = x, y, z$) into

$$\boxed{\tilde{\mathbf{H}}_F = J\sum_{\ell=1}^{L}{}' \Big( \boldsymbol{\sigma}_\ell^x \frac{1-\boldsymbol{\sigma}_{\ell+1}^z}{2}\boldsymbol{\sigma}_{\ell+2}^x + \boldsymbol{\sigma}_\ell^y\frac{1-\boldsymbol{\sigma}_{\ell+1}^z}{2}\boldsymbol{\sigma}_{\ell+2}^y \Big),} \tag{36}$$

where we introduced the notation

$$\sum_{\ell=1}^{L}{}' \mathbf{O}_\ell := \frac{1+\mathbf{\Pi}^z}{2}\Big( \sum_{\substack{\ell=1 \\ \text{PBC}}}^{L} \mathbf{O}_\ell \Big)\frac{1+\mathbf{\Pi}^z}{2} + \frac{1-\mathbf{\Pi}^z}{2}\boldsymbol{\sigma}_L^x\Big( \sum_{\substack{\ell=1 \\ \text{aPBC}}}^{L} \mathbf{O}_\ell \Big)\boldsymbol{\sigma}_L^x\frac{1-\mathbf{\Pi}^z}{2}, \tag{37}$$

with $\mathbf{\Pi}^z = \prod_{\ell=1}^{L}\boldsymbol{\sigma}_\ell^z$. PBC and aPBC refer to periodic and antiperiodic boundary conditions, respectively ($\boldsymbol{\sigma}_{L+n}^{x,y} = \pm\boldsymbol{\sigma}_n^{x,y}$). The operators $\mathbf{F}$ and $\mathbf{H}_I$ are instead mapped into

$$\tilde{\mathbf{F}} = 2J\sum_{\ell=1}^{L}{}' \boldsymbol{\sigma}_\ell^+\frac{1-\boldsymbol{\sigma}_{\ell+1}^z}{2}\boldsymbol{\sigma}_{\ell+2}^+, \qquad \tilde{\mathbf{H}}_I = \frac{J}{4}\sum_{\ell=1}^{L}{}'\boldsymbol{\sigma}_\ell^z = \frac{J}{4}(-i)^{L-1}e^{i\frac{\pi}{2}\sum_{n=1}^{L-1}\boldsymbol{\sigma}_n^z} + \frac{J}{4}\sum_{\ell=1}^{L-1}\boldsymbol{\sigma}_\ell^z. \tag{38}$$

Regarding the other local charges, we conjecture that the transformation $\mathbf{O} \mapsto \tilde{\mathbf{O}}$ reduces the ranges of local densities $\mathbf{q}_{n,\ell}^+$ by one site and leaves the ranges of $\mathbf{q}_{n,\ell}^-$ intact. Thus, $\tilde{\mathbf{q}}_{n,\ell}^\pm$ both act on the same number of sites. Numerical calculation confirms our conjecture for charges with ranges of up to six sites. We also note that $\frac{2}{J}\mathbf{H}_I^-$, given in Eq. (29), is mapped into the staggered spin along the $z$-axis

$$\frac{2}{J}\tilde{\mathbf{H}}_I^- = \mathbf{S}_-^z = \frac{1}{2}\sum_{\ell=1}^{L}{}'(-1)^\ell\boldsymbol{\sigma}_\ell^z. \tag{39}$$

To diagonalise the dual folded Hamiltonian $\tilde{\mathbf{H}}_F$, it is now enough to consider the operator

$$\boxed{\tilde{\mathbf{H}}_F^\eta := J\sum_{\substack{\ell=1 \\ \boldsymbol{\sigma}_{L+n}^{x,y}=(-1)^\eta\boldsymbol{\sigma}_n^{x,y}}}^{L} \big( \boldsymbol{\sigma}_\ell^x\boldsymbol{\sigma}_{\ell+2}^x + \boldsymbol{\sigma}_\ell^y\boldsymbol{\sigma}_{\ell+2}^y \big)\frac{1-\boldsymbol{\sigma}_{\ell+1}^z}{2},} \tag{40}$$

---

[4]Combining the two steps, the transformation reads

$$\boldsymbol{\sigma}_\ell^x \mapsto \begin{cases} -\boldsymbol{\sigma}_1^y\prod_{j=2}^{L-1}\boldsymbol{\sigma}_j^z\boldsymbol{\sigma}_L^y, & \ell = 1, \\ \boldsymbol{\sigma}_{\ell-1}^x\boldsymbol{\sigma}_\ell^x, & \ell > 1, \end{cases} \qquad \boldsymbol{\sigma}_\ell^y \mapsto \begin{cases} \boldsymbol{\sigma}_1^x, & \ell = 1, \\ \boldsymbol{\sigma}_{\ell-1}^x\boldsymbol{\sigma}_\ell^y\prod_{j=\ell+1}^{L-1}\boldsymbol{\sigma}_j^z\boldsymbol{\sigma}_L^y, & \ell > 1, \end{cases} \qquad \boldsymbol{\sigma}_\ell^z \mapsto \prod_{j=\ell}^{L-1}\boldsymbol{\sigma}_j^z\boldsymbol{\sigma}_L^y.$$

which commutes with the total magnetisation $\mathbf{S}^z$ along the $z$-axis[5]. Specifically, we have the following correspondence between the eigenstates of $\tilde{\mathbf{H}}_F$ and those of $\tilde{\mathbf{H}}_F^\eta$:

$$
\left.\begin{aligned}
\tilde{\mathbf{H}}_F\,|\Psi\rangle &= E\,|\Psi\rangle \\
\mathbf{\Pi}^z\,|\Psi\rangle &= |\Psi\rangle
\end{aligned}\right\} \implies \tilde{\mathbf{H}}_F^0\,|\Psi\rangle = E\,|\Psi\rangle\,,
$$
$$
\left.\begin{aligned}
\tilde{\mathbf{H}}_F\,|\Psi\rangle &= E\,|\Psi\rangle \\
\mathbf{\Pi}^z\,|\Psi\rangle &= -|\Psi\rangle
\end{aligned}\right\} \implies \tilde{\mathbf{H}}_F^1\boldsymbol{\sigma}_L^x\,|\Psi\rangle = E\boldsymbol{\sigma}_L^x\,|\Psi\rangle\,.
\tag{41}
$$

Considering this mapping in the opposite direction we note that, in both cases, only the eigenstates of $\tilde{\mathbf{H}}_F^\eta$ that belong to the sector $\mathbf{\Pi}^z = 1$ are needed to diagonalise $\tilde{\mathbf{H}}_F$. In particular, for $\eta = 1$ this is due to the additional action of $\boldsymbol{\sigma}_L^x$ that flips the spin on the last site of the chain and therefore maps the state from one sector to the other. In light of correspondence (41) we will restrict our considerations to $\tilde{\mathbf{H}}_F^\eta$ in the rest of this work.

**Dual XXZ Hamiltonian**

For the sake of completeness, we also report the dual XXZ Hamiltonian, which reads

$$
\tilde{\mathbf{H}} = \sum_{\ell=1}^{L}{}' \boldsymbol{\sigma}_\ell^x(1 - \boldsymbol{\sigma}_{\ell+1}^z)\boldsymbol{\sigma}_{\ell+2}^x + \Delta\boldsymbol{\sigma}_\ell^z.
\tag{42}
$$

This operator commutes with $\mathbf{\Pi}^z$, indeed each of its terms flips two next-nearest neighbour spins simultaneously. On the other hand, the total magnetisation in the direction of the anisotropy is mapped into the operator

$$
\tilde{\mathbf{S}}^z = \frac{1}{2}\sum_{\ell=1}^{L}\prod_{j=\ell}^{L-1}\boldsymbol{\sigma}_j^z\boldsymbol{\sigma}_L^y\,,
\tag{43}
$$

which anti-commutes with $\mathbf{\Pi}^z$. As a consequence, the eigenstates of $\tilde{\mathbf{H}}$ in the sectors $\mathbf{\Pi}^z = \pm 1$ are also eigenstates of $\tilde{\mathbf{S}}^z$ *only if* they are in the kernel of the latter operator. Otherwise, they are degenerate between the two sectors. The same holds for the eigenstates of $\tilde{\mathbf{H}}_F$, and this is arguably the biggest difference between the standard basis of eigenstates and the basis that is implicitly chosen when diagonalising the folded Hamiltonian through $\tilde{\mathbf{H}}_F^\eta$ (or the XXZ Hamiltonian through Eq. (42)).

**The action of $\tilde{\mathbf{H}}_F^\eta$ in the standard spin basis**

Since the total magnetisation along the $z$-axis $\mathbf{S}^z = \frac{1}{2}\sum_\ell \boldsymbol{\sigma}_\ell^z$ commutes with $\tilde{\mathbf{H}}_F^\eta$, there are bases in which the eigenstates of $\tilde{\mathbf{H}}_F^\eta$ are also eigenstates of $\mathbf{S}^z$. We denote by $|\underline{s}\rangle = |s_1,\dots,s_L\rangle$ the eigenstate of the projections of all local spins on the $z$-axis, so that $\boldsymbol{\sigma}_\ell^z|\underline{s}\rangle = s_\ell|\underline{s}\rangle$, for $\ell = 1, 2, \dots, L$. Understanding the action of $\tilde{\mathbf{H}}_F^\eta$ on such states will later be necessary for the identification of the Bethe states that diagonalise the model and the conservation laws. We have

$$
\tilde{\mathbf{H}}_F^\eta\,|\underline{s}\rangle = \sum_{\ell=1}^{L}\tilde{\mathbf{H}}_{F,\ell}^\eta\,|\underline{s}\rangle = J\sum_{\substack{\ell=1 \\ \boldsymbol{\sigma}_{L+n}^x=(-1)^\eta\boldsymbol{\sigma}_n^x}}^{L}\frac{(1-s_{\ell+1})(1-s_\ell s_{\ell+2})}{2}\boldsymbol{\sigma}_\ell^x\boldsymbol{\sigma}_{\ell+2}^x\,|\underline{s}\rangle\,,
\tag{44}
$$

---

[5]Note that the full asymptotic folded Hamiltonian $\tilde{\mathbf{H}}_F$ instead commutes with $\tilde{\mathbf{H}}_I$ from Eq. (38).

where $\tilde{\mathbf{H}}_{F,\ell}^{\eta}$ denotes the local density of $\tilde{\mathbf{H}}_F^{\eta}$, acting on sites $\ell$, $\ell+1$ and $\ell+2$, with boundary condition $\boldsymbol{\sigma}_{L+n}^x = (-1)^{\eta}\boldsymbol{\sigma}_n^x$. The local action of $\tilde{\mathbf{H}}_F^{\eta}$ amounts to

$$
\begin{aligned}
\tilde{\mathbf{H}}_{F,\ell}^{\eta}|\cdots \underset{\ell}{\uparrow\downarrow\downarrow} \cdots\rangle &= 2J[1 - 2\eta(\delta_{\ell,L-1} + \delta_{\ell,L})]|\cdots \underset{\ell}{\downarrow\downarrow\uparrow} \cdots\rangle, \\
\tilde{\mathbf{H}}_{F,\ell}^{\eta}|\cdots \underset{\ell}{\downarrow\downarrow\uparrow} \cdots\rangle &= 2J[1 - 2\eta(\delta_{\ell,L-1} + \delta_{\ell,L})]|\cdots \underset{\ell}{\uparrow\downarrow\downarrow} \cdots\rangle,
\end{aligned}
\tag{45}
$$

where $\downarrow$ stands for $s = -1$ and $\uparrow$ for $s = 1$. The other combinations of three neighbouring spins are destroyed. In words, two oppositely aligned next-nearest neighbour spins are exchanged if the spin between them points downwards.

# 4 Invariants and jamming

## 4.1 Elementary particles

Given that the local density $\tilde{\mathbf{H}}_{F,\ell}^{\eta}$ of $\tilde{\mathbf{H}}_F^{\eta}$ is supported on three neighbouring sites labeled by $\ell, \ell+1, \ell+2$ (see, for example, Eq. (40)), the interaction becomes of the nearest-neighbour type if we define *macrosites* that consist of two neighbouring lattice sites. We use the following notation:

$$
|\cdots\varnothing\cdots\rangle := |\cdots\downarrow\downarrow\cdots\rangle, \quad |\cdots\ominus\cdots\rangle := |\cdots\uparrow\downarrow\cdots\rangle, \quad |\cdots\oplus\cdots\rangle := |\cdots\downarrow\uparrow\cdots\rangle, \quad |\cdots\underline{\ominus\oplus}\cdots\rangle := |\cdots\uparrow\uparrow\cdots\rangle.
\tag{46}
$$

Clearly there is a local ambiguity in the position of the macrosite, which could be defined to start either from an odd or from an even site. If the chain has an even number $L$ of sites, this ambiguity is irrelevant, as it corresponds to fixing a convention. If instead $L$ is odd, the two options are linked by the boundary. We will see later how this can be taken into account; for the moment the reader can assume that the chain has an even number of sites. The macrosite will be denoted by a primed index $\ell'$ and consists of the neighbouring sites $2\ell'-1$ and $2\ell'$.

The dynamics generated by $\tilde{\mathbf{H}}_F^{\eta}$ can be summarised by the following actions of local densities:

$$
\begin{aligned}
\tilde{\mathbf{H}}_{F,2\ell'}^{\eta}|\cdots\underset{\ell'}{\varnothing}\underset{\ell'+1}{\oplus}\cdots\rangle &= 2J|\cdots\underset{\ell'}{\oplus}\underset{\ell'+1}{\varnothing}\cdots\rangle, & \tilde{\mathbf{H}}_{F,2\ell'}^{\eta}|\cdots\underset{\ell'}{\oplus}\underset{\ell'+1}{\varnothing}\cdots\rangle &= 2J|\cdots\underset{\ell'}{\varnothing}\underset{\ell'+1}{\oplus}\cdots\rangle, \\
\tilde{\mathbf{H}}_{F,2\ell'-1}^{\eta}|\cdots\underset{\ell'}{\varnothing}\underset{\ell'+1}{\ominus}\cdots\rangle &= 2J|\cdots\underset{\ell'}{\ominus}\underset{\ell'+1}{\varnothing}\cdots\rangle, & \tilde{\mathbf{H}}_{F,2\ell'-1}^{\eta}|\cdots\underset{\ell'}{\ominus}\underset{\ell'+1}{\varnothing}\cdots\rangle &= 2J|\cdots\underset{\ell'}{\varnothing}\underset{\ell'+1}{\ominus}\cdots\rangle, \\
\tilde{\mathbf{H}}_{F,2\ell'-1}^{\eta}|\cdots\underset{\ell'}{\ominus}\underset{\ell'+1}{\oplus}\cdots\rangle &= 2J|\cdots\underset{\ell'}{\varnothing}\underset{\ell'+1}{\underline{\ominus\oplus}}\cdots\rangle, & \tilde{\mathbf{H}}_{F,2\ell'-1}^{\eta}|\cdots\underset{\ell'}{\varnothing}\underset{\ell'+1}{\underline{\ominus\oplus}}\cdots\rangle &= 2J|\cdots\underset{\ell'}{\ominus}\underset{\ell'+1}{\oplus}\cdots\rangle, \\
\tilde{\mathbf{H}}_{F,2\ell'}^{\eta}|\cdots\underset{\ell'}{\ominus}\underset{\ell'+1}{\oplus}\cdots\rangle &= 2J|\cdots\underset{\ell'}{\underline{\ominus\oplus}}\underset{\ell'+1}{\varnothing}\cdots\rangle, & \tilde{\mathbf{H}}_{F,2\ell'}^{\eta}|\cdots\underset{\ell'}{\underline{\ominus\oplus}}\underset{\ell'+1}{\varnothing}\cdots\rangle &= 2J|\cdots\underset{\ell'}{\ominus}\underset{\ell'+1}{\oplus}\cdots\rangle.
\end{aligned}
\tag{47}
$$

The remaining (orthogonal) cases of occupations of the macrosites $\ell'$ and $\ell'+1$ are sent to zero by the local densities $\tilde{\mathbf{H}}_{F,2\ell'-1}^{\eta}$ and $\tilde{\mathbf{H}}_{F,2\ell'}^{\eta}$. In particular, we see that the state with all spins down (all macrosites are occupied by $\varnothing$) is an eigenstate of $\tilde{\mathbf{H}}_F^{\eta}$ with zero eigenvalue. The first four rules in Eq. (47) then suggest to identify such a state with the vacuum

$$
|\phi\rangle := |\downarrow\downarrow\cdots\downarrow\rangle
\tag{48}
$$

of two species of particles, $\oplus$ and $\ominus$, that move freely until another particle is met; their interaction is then encoded in the remaining four rules of Eq. (47). Within this picture, the

Figure 1: States forming a basis for the configurations $\{(1,1,0,1,0)\}_c$ (a) and $\{(1,0,0,1,0)\}_c$ (b) with $L = 8$. The representation in terms of elementary particles is displayed close to the state. Note that the two invariant spaces are mapped into one another under a shift by one site (the effect of this shift is the replacement $b_j \rightarrow 1 - b_j$).

state $\ominus\oplus$ can be interpreted as a particle $\ominus$ and a particle $\oplus$ sharing the same macrosite. Note that a particle of one type cannot jump across a particle of the other, thus *the domains of particles $\oplus$ and $\ominus$ do not change under time evolution*. Going back to the folded Hamiltonian, we note that the vacuum state $|\phi\rangle$ is associated with an eigenstate of $\tilde{\mathbf{H}}_F$ only if the size $L$ of the system is even (condition $\mathbf{\Pi}^z |\phi\rangle = 1$ should be fulfilled – see Eq. (41)).

**Noninteracting sectors**

Let us consider a state with a single species of particles, say $\oplus$. The entire dynamics is described by the first two rules in Eq. (47). If $L$ is even, the mapping $\oplus \mapsto \uparrow$, $\varnothing \mapsto \downarrow$ sends $\tilde{\mathbf{H}}_F^\eta$ into the Hamiltonian of the XX model with $L/2$ spins,

$$\mathbf{H}_{\mathrm{XX}}^\eta = J \sum_{\substack{\ell=1 \\ \boldsymbol{\sigma}_{L/2+1}^{x,y} = (-1)^\eta \boldsymbol{\sigma}_1^{x,y}}}^{L/2} \mathbf{K}_{\ell,\ell+1} \,. \tag{49}$$

This Hamiltonian describes a noninteracting spin chain. For $(-1)^\eta = (-1)^{L/2}$, $\mathbf{H}_{\mathrm{XX}}^\eta$ possesses a non-abelian set of local conservation laws. Since both boundary conditions are compatible with the existence of non-commuting charges, this sector of $\mathbf{H}_F$ has the phenomenology typical of non-abelian integrable systems, as first discussed in Ref. [15].

For $L$ odd, the hypothesis of having a single species of particles is meaningless: the two species are transformed one into the other when crossing the boundary. The closest situation is having no more than two groups of particles of different species. Such configurations would still form an invariant subspace, like in the even case, but now in presence of interaction. We postpone the analysis of such a case to the next sections.

## 4.2 Topological invariants

If we indicate by $\ell'_j$ the macrosite of the $j$-th particle, the actual position of the corresponding spin up is $\ell_j = 2\ell'_j - b_j$, where $b_j = 0$ for a particle of type $\oplus$ and $b_j = 1$ for a particle of type $\ominus$. Assuming $L$ to be even, rules (47) imply that the set $B_N^{(\mathrm{e})} = \{(b_1, \ldots, b_N)\}_c$ of *different* cyclic permutations of the sequence $(b_1, \ldots, b_N)$, where $N$ is the number of spins up, is preserved by the Hamiltonian: particles of different type cannot jump across each

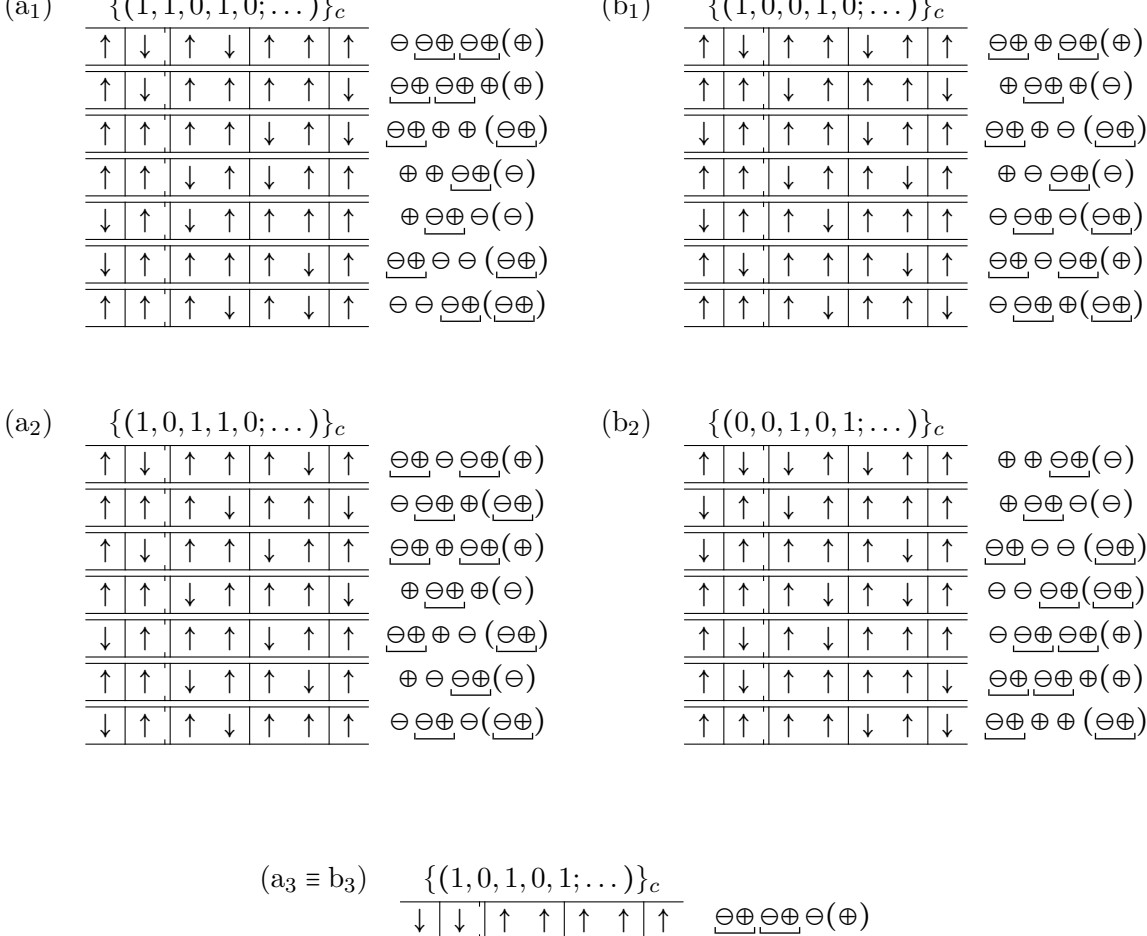

Figure 2:    The same as in Fig. 1 for an odd chain with $L = 7$ sites; we report all the configurations analogous to those in Fig. 1 (there are more options because the sequences in Fig. 1 are defined up to cyclic permutations). The particle in parentheses is transmuted due to the periodic boundary conditions (dashed vertical line). Note that in odd chains each configuration represents a space that is invariant under a shift by one site. We point out that the states in ($a_1$), ($a_2$), ($b_1$) and ($b_2$) are jammed (indeed the particles can not move), so each one separately generates an invariant subspace.

other – see Fig. 1. This invariance is not present in the original XXZ Hamiltonian, so it is a candidate for generating pre-relaxational behaviour at large anisotropy. Even if $L$ is odd a similar result holds, however, in order to account for the transmutation of particles at the boundary, the configuration of particles must be encoded in a larger set of sequences, namely $B_N^{(o)} = \{(b_1, \ldots, b_N; 1 - b_1, \ldots, 1 - b_N)\}_c$ – see Fig. 2.[6] As a result, contrary to the even-size case, the space associated with a configuration is invariant under a translation by one site.

In general, the configuration is a "topological invariant" in the sense that it is independent of the actual positions of the particles and hence does not change when the particles are arbitrarily moved without crossing.

We point out that all functionals of the set $B_N^{(e)}$ (or $B_N^{(o)}$) are associated with a diagonal conserved operator. Apart from $\frac{1}{2} \sum_{\ell'=1}^{L/2} (\mathbf{1} + \boldsymbol{\sigma}_{2\ell'-1}^z)$, which has eigenvalues $\sum_{j=1}^N b_j$, such charges are not local, and it is not evident whether they could be defined as quasilocal [33], i.e., as translationally invariant sums of local densities that are exponentially suppressed in their range[7]. Assuming $L$ to be even, an important example is provided by the following charge

$$\mathbf{M} = \sum_{\ell'=1}^{\frac{L}{2}} \sum_{n'=0}^{\frac{L}{2}-1} \frac{\mathbf{1} + \boldsymbol{\sigma}_{2\ell'-1}^z}{2} \left( \prod_{j=2\ell'}^{2\ell'-1+2n'} \frac{\mathbf{1} - \boldsymbol{\sigma}_j^z}{2} \right) \frac{\mathbf{1} + \boldsymbol{\sigma}_{2\ell'+2n'}^z}{2}, \tag{50}$$

whose eigenspaces are characterised by the configurations $B_N^{(e)}$, and which has eigenvalues

$$\boxed{M = \sum_{\substack{j=1 \\ b_{N+1}=b_1}}^{N} b_j (1 - b_{j+1}).} \tag{51}$$

These eigenvalues count the number of *oriented domain walls* separating particles of type $\ominus$ on the left-hand side from those of type $\oplus$ on the right-hand side. We point out that $\mathbf{M}$ is quasilocal, in particular, it is extensive, in the sense that

$$\lim_{L \to \infty} \frac{1}{L} \left( \langle \mathbf{M}^2 \rangle - \langle \mathbf{M} \rangle^2 \right) < \infty, \tag{52}$$

where $\langle \bullet \rangle$ denotes the expectation value in some physically relevant state with a finite correlation length, e.g., a thermal state, or a generalised Gibbs ensemble [34]. Such states will be considered in the second part of our work [35], where we elaborate on the thermodynamic limit of the Bethe Ansatz.

The importance of $\mathbf{M}$ will become evident in Section 5, here we only mention its appearance in a necessary and sufficient condition for the existence of the configuration $B_N^{(e)}$: $L \geq 2(N - M)$. This is a consequence of the fact that generally each particle corresponds to a pair $\uparrow\downarrow$ or $\downarrow\uparrow$, for a total of $2N$ sites; exceptions, however, can arise on the $M$ oriented domain

---

[6]Note that the configuration $(b_1, \ldots, b_N; 1 - b_1, \ldots, 1 - b_N)$ describes the particle content of two side-by-side copies of the spin configuration, i.e., $(s_1, \ldots, s_N; s_1, \ldots, s_N)$.

[7]The range of a localised operator is the number of sites on which the operator acts nontrivially. By exponential localisation we mean that the local density can be written as a sum of operators that have distinct ranges, and whose fluctuations in a generic macro-state decay exponentially with their range. For example, in equilibrium at infinite temperature the fluctuation of a projector $\boldsymbol{\Pi}_n$ acting on $n$ adjacent sites reads

$$\langle \boldsymbol{\Pi}_n^2 \rangle - \langle \boldsymbol{\Pi}_n \rangle^2 = \langle \boldsymbol{\Pi}_n \rangle (1 - \langle \boldsymbol{\Pi}_n \rangle) = \frac{\operatorname{tr} \boldsymbol{\Pi}_n}{\operatorname{tr} \mathbf{1}} \left( 1 - \frac{\operatorname{tr} \boldsymbol{\Pi}_n}{\operatorname{tr} \mathbf{1}} \right) = 2^{-n}(1 - 2^{-n}).$$

walls, where $\ominus$ and $\oplus$ can occupy a single macrosite (we then denote them by $\underline{\ominus\oplus}$), releasing $2M$ sites.

Incidentally, we note that the average distance between two particles, measured in the number of macrosites, is $\xi = \frac{L}{2N}$. Since $L \geq 2(N - M)$, $\xi$ can not be smaller than $1 - \mu$, where $\mu = M/N$ parametrises the average degree of macro-degeneracy of particles. The configurations with minimal average distance between particles, namely, the ones with $L = 2(N - M)$, are special and will be investigated in detail in the next section.

## 4.3 Jammed states

In this section we study the subspace with $L = 2(N-M)$, which is the ground-state eigenspace of the quasilocal charge $\mathbf{M} - \mathbf{S}^z$. Without making assumptions on the behaviour of the states under translations, we exhibit an orthonormal set of stationary states spanning the entire subspace. We nonetheless anticipate that the Bethe Ansatz solution that will be worked out in Section 5 is complete, including also such states in the form of linear combinations invariant under shifts by two sites.

In the states with $L = 2(N - M)$ a spin down is always surrounded by spins up, and from Eq. (45) it follows that any product state $|\Psi\rangle$ with that property belongs to the kernel of $\tilde{\mathbf{H}}_F^\eta$. Such a condition can be expressed as

$$\sum_{\ell=1}^{L} \frac{\mathbf{1} - \boldsymbol{\sigma}_\ell^z}{2} \frac{\mathbf{1} - \boldsymbol{\sigma}_{\ell+1}^z}{2} |\Psi\rangle = 0, \tag{53}$$

that is to say, $|\Psi\rangle$ is in the eigenspace of the ground state of the classical Ising model

$$\mathbf{H}_{\mathrm{Is}} = \sum_{\ell=1}^{L} \frac{1}{4}\boldsymbol{\sigma}_\ell^z\boldsymbol{\sigma}_{\ell+1}^z - \frac{1}{2}\boldsymbol{\sigma}_\ell^z. \tag{54}$$

Each product state corresponds to a configuration where the particles ($\oplus$ and $\ominus$) are jammed – see Fig. 2; for that reason we call them *jammed states*. Remarkably, translational invariance can be broken in a chaotic way, as the jammed states consist of strings of spins up with arbitrary length, interspersed with single spins down. As a result, jammed states can play a key role in nonequilibrium time evolution, preventing restoration of one- or two-site shift invariance when the initial state is not symmetric.

The connection with the classical Ising model allows us to easily compute the size of the space

$$d_{\mathrm{jam}} = \lim_{\beta\to\infty} \mathrm{tr}\left[e^{-\beta\mathbf{H}_{\mathrm{Is}}}\right] e^{-\frac{\beta L}{4}} = \left(\frac{1 + \sqrt{5}}{2}\right)^L \approx 1.618^L, \tag{55}$$

which is, remarkably, exponentially large.[8] The number of jammed states at fixed number $N$ of spins up is instead given by

$$\binom{N}{L - N} + \binom{N - 1}{L - N - 1} = \frac{L}{N}\binom{N}{L - N}. \tag{56}$$

---

[8]Note that the dimension of the projected Hilbert space, where the constrained Hamiltonian (32) acts, scales in the same way. Indeed, in the original (non-compressed) basis the projector (33) destroys the states that contain two consecutive spins down.

The first term counts the product states in which the left-most position is occupied by a spin up. Since two neighbouring spins cannot point downwards, spins down should always be paired with a spin up. We therefore have to distribute $N$ "fragments" of consecutive spins, of which $L - N$ are of type $\uparrow\downarrow$ and the rest of type $\uparrow$. In contrast, the second term in Eq. (56) counts the configurations in which the left-most position is occupied by a spin down and the right-most one by a spin up (the latter can not be down because of periodic boundary conditions). We have then to distribute $N - 1$ fragments, of which $L - N - 1$ are of type $\uparrow\downarrow$, the rest being spins up. Incidentally, we note that, since every spin down comes in a pair with a spin up, the total magnetisation of a jammed product state is always non-negative, i.e., $S^z = \langle \mathbf{S}^z \rangle = N - \frac{L}{2} \geq 0$.

Transforming back to the folded Hamiltonian (27), only the jammed states with an even number of spins down, i.e., those belonging to the sector $\mathbf{\Pi}^z = 1$, correspond to eigenstates of $\mathbf{H}_F$. Indeed, according to Eq. (41), if $|\underline{s}\rangle$ is a jammed state of $\tilde{\mathbf{H}}_F^\eta$ with $\prod_{j=1}^L s_j = 1$, the corresponding stationary state of $\tilde{\mathbf{H}}_F$ is $|s_1, \ldots, s_{L-1}, (-1)^\eta s_L\rangle$, belonging in turn to the sector $\mathbf{\Pi}^z = (-1)^\eta$. The corresponding stationary state of $\mathbf{H}_F$ is then

$$\frac{1}{\sqrt{2}} \sum_{s=\pm 1} s^\eta |s, ss_1, ss_1 s_2, \ldots, ss_1 \cdots s_{L-1}\rangle, \tag{57}$$

as one can easily check by applying the inverse of the duality transformation, mapping $\tilde{\mathbf{H}}_F$ into $\mathbf{H}_F$:

$$\boldsymbol{\sigma}_\ell^z \mapsto \begin{cases} \boldsymbol{\sigma}_\ell^z \boldsymbol{\sigma}_{\ell+1}^z, & \ell < L, \\ -\boldsymbol{\sigma}_1^y \prod_{j=2}^{L-1} \boldsymbol{\sigma}_j^x \boldsymbol{\sigma}_L^y, & \ell = L. \end{cases} \tag{58}$$

Finally, a symmetric and antisymmetric combination of the states (57) with $\eta = 0$ and $\eta = 1$ produces the jammed product (eigen)states of $\mathbf{H}_F$, namely

$$|\pm 1, \pm s_1, \pm s_1 s_2, \ldots, \pm s_1 \cdots s_{L-1}\rangle. \tag{59}$$

These are the product states in which all the domains consisting of spins aligned in the same direction have length larger than 1. This follows from Eq. (53), which prohibits two consecutive $s_\ell$ (in the dual basis) from being equal to $-1$. Interestingly, cluster decomposition properties, which were absent in the state shown in Eq. (57), are restored by mixing jammed states belonging to different sectors of the dual folded Hamiltonian $\tilde{\mathbf{H}}_F$.

**Jammed states under the leading correction**

The leading asymptotic correction to the dual folded Hamiltonian reads $\tilde{\mathbf{H}}_F' = J^{-1}[\tilde{\mathbf{F}}, \tilde{\mathbf{F}}^\dagger]$, where $\tilde{\mathbf{F}}$ is given in Eq. (38) (see also Eq. (9)). Explicitly, we have

$$\tilde{\mathbf{H}}_F' = 4J \sum_{\ell=1}^L {}' \boldsymbol{\sigma}_\ell^z \frac{1 - \boldsymbol{\sigma}_{\ell-1}^z}{2} \frac{1 - \boldsymbol{\sigma}_{\ell+1}^z}{2} (\boldsymbol{\sigma}_{\ell-2}^+ \boldsymbol{\sigma}_{\ell+2}^- + h.c.) + \boldsymbol{\sigma}_\ell^z \frac{1 - \boldsymbol{\sigma}_{\ell+1}^z}{2} + (\boldsymbol{\sigma}_{\ell-1}^+ \boldsymbol{\sigma}_\ell^- \boldsymbol{\sigma}_{\ell+1}^+ \boldsymbol{\sigma}_{\ell+2}^- + h.c.), \tag{60}$$

which can be rewritten as $\tilde{\mathbf{H}}_F' = 8\tilde{\mathbf{H}}_I - 2\tilde{\mathbf{Q}}_2^+ + J \sum_\ell'(\mathbf{D}_{\ell,\ell+3}\mathbf{D}_{\ell+1,\ell+2} - 2\boldsymbol{\sigma}_\ell^z \boldsymbol{\sigma}_{\ell+1}^z)$, where $\tilde{\mathbf{H}}_I$ (reported in Eq. (38)) and $\tilde{\mathbf{Q}}_2^{+}$[9] are the dual conserved charges $\mathbf{H}_I$ and $\mathbf{Q}_2^+$, whose densities are given in Eqs. (25) and (31), respectively.

---
[9]

$$\tilde{\mathbf{Q}}_2^+ = \frac{J}{4} \sum_{\ell=1}^L {}' \mathbf{D}_{\ell,\ell+3}\mathbf{D}_{\ell+1,\ell+2} - \mathbf{K}_{\ell,\ell+3}\mathbf{K}_{\ell+1,\ell+2} - (1 - \boldsymbol{\sigma}_{\ell+1}^z)\boldsymbol{\sigma}_{\ell+2}^z(1 - \boldsymbol{\sigma}_{\ell+3}^z)\mathbf{K}_{\ell,\ell+4}.$$

The leading correction spoils the conservation of the configuration and, in turn, the jamming, as evident in the following actions of its local density:

$$
\begin{aligned}
\tilde{\mathbf{H}}'_{F,\ell}|\cdots \underset{\ell}{\uparrow\downarrow\uparrow\downarrow\uparrow} \cdots\rangle &= 4J|\cdots \downarrow\uparrow\downarrow\uparrow\uparrow \cdots\rangle + 4J|\cdots \uparrow\downarrow\uparrow\downarrow\uparrow \cdots\rangle, \\
\tilde{\mathbf{H}}'_{F,\ell}|\cdots \underset{\ell}{\downarrow\uparrow\downarrow\uparrow\downarrow} \cdots\rangle &= 4J|\cdots \uparrow\downarrow\uparrow\downarrow\downarrow \cdots\rangle.
\end{aligned}
\tag{61}
$$

The subset of states $|\Psi\rangle$ that remain jammed even when the leading correction $\tilde{\mathbf{H}}'_F$ is included has an additional property: no substrings of the form $\downarrow\uparrow\downarrow$ are present in the spin configuration of $|\Psi\rangle$. This can be compactly expressed as follows:

$$
\sum_{\ell=1}^{L}\left(\frac{1-\sigma_{\ell}^z}{2} + \frac{1-\sigma_{\ell-1}^z}{2}\frac{1+\sigma_{\ell}^z}{2}\right)\frac{1-\sigma_{\ell+1}^z}{2}|\Psi\rangle = 0.
\tag{62}
$$

Since each spin down has to be surrounded by distinct spins up, the minimal number of spins up is $\lceil\frac{2}{3}L\rceil$; the total magnetisation is therefore bounded from below by $S^z \geq \lceil\frac{1}{6}L\rceil$.

We note that the energy of the jammed states is not zero anymore – it is now given by

$$
\langle\Psi|\tilde{\mathbf{H}}_F + \kappa\tilde{\mathbf{H}}'_F|\Psi\rangle = 4J\kappa(L-N).
\tag{63}
$$

Since all the energies are multiples of $4J\kappa$, the dynamics in the subspace of the jammed states is periodic with period $\pi/(2J\kappa)$.

For a given $N$, and hence energy, the number of jammed product states reads

$$
\binom{2N-L}{L-N} + 2\binom{2N-L-1}{L-N-1} = \frac{L}{2N-L}\binom{2N-L}{L-N}.
\tag{64}
$$

To see this, let us use that a spin down should always be surrounded by spins up, while a spin up can stand alone. The first term accounts for the configurations starting and ending with a spin up. It counts the number of ways in which $L-N$ fragments of type $\uparrow\downarrow\uparrow$ can be distributed among the stand-alone spins up, $\uparrow$. The total number of fragments $\uparrow\downarrow\uparrow$ and $\uparrow$ to distribute is $2N-L$. Conversely, the second term counts the configurations that either start or end with a spin down. There, one has to count the number of distributions of $L-N-1$ fragments of type $\uparrow\downarrow\uparrow$ among the stand-alone spins up, fixing $\downarrow\uparrow$ on the first two sites and $\uparrow$ on the last site (or vice versa, whence the prefactor 2). As in the absence of the leading correction, fixing the boundary spins is a consequence of periodic boundary conditions and of the constraint that prohibits substrings of the form $\downarrow\uparrow\downarrow$. The total number of fragments $\uparrow\downarrow\uparrow$ and $\uparrow$ that have to be distributed is now $2N-L-1$.

The total number of jammed states is obtained by summing Eq. (64) over all allowed $N$:

$$
\sum_{N=\lceil\frac{2}{3}L\rceil}^{L}\binom{2N-L}{L-N} + 2\sum_{N=\lceil\frac{2}{3}L\rceil}^{L-1}\binom{2N-L-1}{L-N-1} \sim \chi^L \approx 1.466^L.
\tag{65}
$$

The (leading) asymptotic approximation $\chi^L$, where $\chi < 1$ is the real solution to $\chi^2(\chi-1) = 1$, arises from one of the terms in the first sum. Again, the kernel of the folded Hamiltonian contains an exponentially large number of jammed states, but its size is negligible with respect to the one without the leading correction – $cf.$ Eq. (55). Transforming back to the folded XXZ Hamiltonian, the additional constraint means that the domains consisting of spins aligned in the same direction must contain at least three neighbouring sites.

We point out that, in order to apply this result to the XXZ model in the limit of large anisotropy, a further step should be made: when the leading correction is considered, the second formula of Eq. (3) implies that the actual asymptotically jammed states $|\Psi(0)\rangle$ are obtained by acting on the jammed states of $\mathbf{H}_F(\kappa)$ with operator $e^{-i\kappa\mathbf{B}_1(\kappa)} = e^{(\mathbf{F}^\dagger-\mathbf{F})/(4J\Delta)+\mathcal{O}(\Delta^{-2})}$ – see Eq. (10). They read

$$\sim \exp\left[-\frac{i}{4\Delta}\sum_\ell(\boldsymbol{\sigma}_\ell^x\boldsymbol{\sigma}_{\ell+1}^y - \boldsymbol{\sigma}_\ell^y\boldsymbol{\sigma}_{\ell+1}^x)\frac{\boldsymbol{\sigma}_{\ell+2}^z - \boldsymbol{\sigma}_{\ell-1}^z}{2}\right]|\pm 1, \pm s_1, \pm s_1 s_2, \ldots, \pm s_1\cdots s_{L-1}\rangle, \qquad (66)$$

where $s_\ell$ satisfy

$$\prod_{\ell=1}^L s_\ell = 1, \quad s_{\ell-1} = -1 \Rightarrow s_\ell = s_{\ell+1} = 1, \qquad \text{and} \qquad s_\ell = 1 \Rightarrow s_{\ell-1} = 1 \vee s_{\ell+1} = 1. \qquad (67)$$

The correction resulting from the unitary transformation in Eq. (66) is exhaustive at $\mathcal{O}(\Delta^{-1})$ only if $\Delta^{-1}Jt \ll 1$. If, instead, $\Delta^{-1}Jt \sim 1$, the next order of the expansion in the folded Hamiltonian, *i.e.*, the leading term in the $\mathcal{O}(\kappa^2)$ remainder in Eq. (9), gives an $\mathcal{O}(\Delta^{-1})$ correction that competes with the one in Eq. (66).

## 5   Coordinate Bethe Ansatz

In this section we diagonalise the asymptotic dual folded Hamiltonian $\tilde{\mathbf{H}}_F$, defined in Section 3.2. Remarkably, the solution that we present is independent of the standard Bethe Ansatz for the XXZ model. This is possible due to three conditions that arise in the strong coupling limit:

1. other reference states can be used as vacua of the Bethe Ansatz description;

2. the particle structure of the XXZ model becomes redundant, in the sense that particles can be reorganised in more efficient ways;

3. the spectrum becomes highly degenerate, and inequivalent bases can be chosen.

The difference with the standard Bethe Ansatz can be recognised a priori. Firstly, the total magnetisation $\mathbf{S}^z$, which counts the number of rapidities associated with the standard Bethe Ansatz eigenstates, is not diagonal in our basis of eigenstates of $\mathbf{H}_F$. This is because its dual $\tilde{\mathbf{S}}^z$ does not preserve the sectors that block diagonalise the dual folded Hamiltonian – see Eqs. (37) and (41), and the discussion after Eq. (43). Secondly, since the particle configuration is conserved, the asymptotic dual folded Hamiltonian $\tilde{\mathbf{H}}_F$ can be diagonalised at fixed configuration. This is an additional structure that is not present in the coordinate Bethe Ansatz of the XXZ model, where one only exploits the conservation of the particle number.

### 5.1   Even number of sites

The eigenstates corresponding to the set $B_N^{(e)} = \{(b_1, \ldots, b_N)\}_c$ of distinct cyclic permutations of the sequence $\underline{b} = (b_1, \ldots, b_N)$ can be represented by a set of $N$ momenta (rapidities)

$\{p_1, \ldots, p_N\}$. The states are of the form

$$|p_1, \ldots, p_N\rangle_{B_N^{(e)}} = \sum_{\underline{b} \in B_N^{(e)}} \sum_{\substack{\ell_1', \ldots, \ell_N' = 1 \\ 2\ell_j' - b_j < 2\ell_{j+1}' - b_{j+1}}}^{\frac{L}{2}} c_{\ell_1', \ldots, \ell_N'}^{b_1, \ldots, b_N}(p_1, \ldots, p_N) \prod_{j=1}^{N} \sigma_{2\ell_j' - b_j}^x |\phi\rangle, \tag{68}$$

in which each term has the same *relative* spatial distribution of particles, enforced by the constraint $2\ell_j' - b_j < 2\ell_{j+1}' - b_{j+1}$ in the sum. The coefficients of the terms that describe a particular spatial configuration $\underline{b}$ of particles read

$$c_{\ell_1', \ldots, \ell_N'}^{b_1, \ldots, b_N}(p_1, \ldots, p_n) = Z_{p_1, \ldots, p_N}^{b_1, \ldots, b_N} \sum_{\pi} S_{b_1, \ldots, b_N}\begin{pmatrix} p_1, \ldots, p_N \\ p_{\pi(1)}, \ldots, p_{\pi(N)} \end{pmatrix} e^{i \sum_{j=1}^{N} \ell_j' p_{\pi(j)}}. \tag{69}$$

Here, $S_{b_1, \ldots, b_N}\begin{pmatrix} p_1, \ldots, p_N \\ p_{\pi(1)}, \ldots, p_{\pi(N)} \end{pmatrix}$ is the multi-particle scattering matrix that describes the permutation $(p_1, \ldots, p_N) \mapsto (p_{\pi(1)}, \ldots, p_{\pi(N)})$ of momenta due to two-particle scattering processes, which leave the relative spatial distribution of particles intact, exchanging only their momenta. In particular, we assume

$$S_{b_1, \ldots, b_N}\begin{pmatrix} p_1, \ldots, p_N \\ p_1, \ldots, p_N \end{pmatrix} = 1, \quad S_{b_1, \ldots, b_N}\begin{pmatrix} p_1, \ldots, p_N \\ p_{\pi(1)}, \ldots, p_{\pi(N)} \end{pmatrix} S_{b_1, \ldots, b_N}\begin{pmatrix} p_{\pi(1)}, \ldots, p_{\pi(N)} \\ p_1, \ldots, p_N \end{pmatrix} = 1, \tag{70}$$

*i.e.*, the amplitude of the process without collisions between particles is 1, and each process is reversible. We remind the reader that the exceptionality of this Ansatz is in preserving the set of momenta under the scattering process.

The boundary conditions $\sigma_{L+n}^x = (-1)^\eta \sigma_n^x$ imply

$$c_{\ell_1', \ldots, \ell_N'}^{b_1, \ldots, b_N}(p_1, \ldots, p_N) = (-1)^\eta c_{\ell_2', \ldots, \ell_n', \ell_1' + \frac{L}{2}}^{b_2, \ldots, b_N, b_1}(p_1, \ldots, p_N), \tag{71}$$

*i.e.*, a prefactor $(-1)^\eta$ appears each time a particle is moved across the boundary from the first to the last position in the configuration. We then find

$$\frac{Z_{p_1, \ldots, p_N}^{b_1, \ldots, b_N}}{Z_{p_1, \ldots, p_N}^{b_2, \ldots, b_N, b_1}} = (-1)^\eta S_{b_1, \ldots, b_N}\begin{pmatrix} p_{\pi(1)}, \ldots, p_{\pi(N)} \\ p_1, \ldots, p_N \end{pmatrix} S_{b_2, \ldots, b_N, b_1}\begin{pmatrix} p_1, \ldots, p_N \\ p_{\pi(2)}, \ldots, p_{\pi(N)}, p_{\pi(1)} \end{pmatrix} e^{i \frac{L}{2} p_{\pi(1)}}, \tag{72}$$

for a generic permutation $\pi$. Note that any ratio $Z_{p_1, \ldots, p_N}^{b_j, \ldots, b_N, b_1, \ldots, b_{j-1}} / Z_{p_1, \ldots, p_N}^{b_{j+1}, \ldots, b_N, b_1, \ldots, b_j}$ satisfies Eq. (72), provided that we shift the configuration $\underline{b}$ in the scattering matrix accordingly (the indices of the momenta should, however, remain the same).

The Bethe equations are now a result of enforcing, (i), the $\pi$-independence of Eq. (72), and, (ii), the conservation of the total momentum. The latter comes from imposing

$$\frac{Z_{p_1, \ldots, p_N}^{b_2, \ldots, b_N, b_1}}{Z_{p_1, \ldots, p_N}^{b_1, \ldots, b_N}} \frac{Z_{p_1, \ldots, p_N}^{b_3, \ldots, b_N, b_1, b_2}}{Z_{p_1, \ldots, p_N}^{b_2, \ldots, b_N, b_1}} \cdots \frac{Z_{p_1, \ldots, p_N}^{b_N, b_1, \ldots, b_{N-1}}}{Z_{p_1, \ldots, p_N}^{b_{N-1}, b_N, b_1 \ldots, b_{N-2}}} \frac{Z_{p_1, \ldots, p_N}^{b_1, \ldots, b_N}}{Z_{p_1, \ldots, p_N}^{b_N, b_1, \ldots, b_{N-1}}} = 1 \tag{73}$$

on the ratios exhibited in Eq. (72) for the sequence of permutations $\pi(j) = j$ in the first ratio, $\pi(j) = j + 1$ in the second, and so on and so forth. The scattering matrices then cancel due to the reversibility of the scattering process (right-hand side of Eq. (70)), and we obtain

$$\boxed{e^{i \frac{L}{2} \sum_{j=1}^{N} p_j} = e^{iN\pi\eta}.} \tag{74}$$

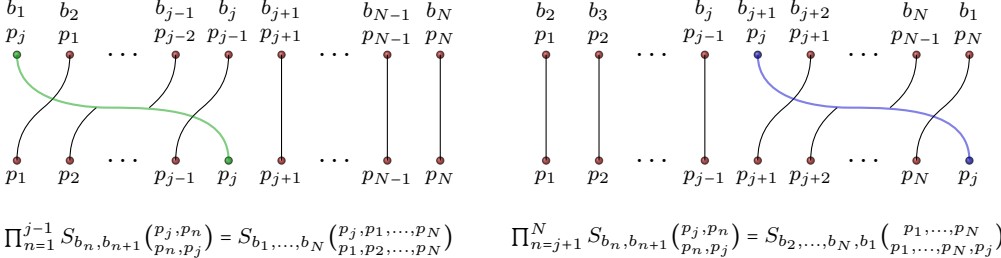

$$\prod_{n=1}^{j-1} S_{b_n,b_{n+1}}\binom{p_j,p_n}{p_n,p_j} = S_{b_1,\ldots,b_N}\binom{p_j,p_1,\ldots,p_N}{p_1,p_2,\ldots,p_N} \qquad \prod_{n=j+1}^{N} S_{b_n,b_{n+1}}\binom{p_j,p_n}{p_n,p_j} = S_{b_2,\ldots,b_N,b_1}\binom{p_1,\ldots,p_N}{p_1,\ldots,p_N,p_j}$$

Figure 3: Factorisation of the scattering processes that enter Eq. (72). The chosen permutation is $\pi : (j, 1, \ldots, N) \mapsto (1, 2, \ldots, N)$. The dots represent particles of types $b_1, \ldots, b_N$, to which momenta are assigned. The lines connecting them represent the exchange of momenta during collisions. Each crossing of two lines contributes a two-particle scattering matrix to the product that, at the end, represents the entire scattering process.

The condition that Eq. (72) should not depend on the choice of permutation can be used to obtain an additional identity. Choosing, for instance, permutation $\pi : (1, 2, \ldots, N) \mapsto (j, 1, \ldots, N)$ on the one hand and $\pi' : (1, 2, \ldots, N) \mapsto (\ell, 1, \ldots, N)$ on the other, we get

$$e^{i\frac{L}{2}(p_\ell - p_j)} = \prod_{\substack{n=1 \\ n \neq j}}^{N} S_{b_n,b_{n+1}}\binom{p_j, p_n}{p_n, p_j} \prod_{\substack{m=1 \\ m \neq \ell}}^{N} S_{b_m,b_{m+1}}\binom{p_m, p_\ell}{p_\ell, p_m}, \tag{75}$$

where, by virtue of integrability, we expressed the multi-particle scattering matrices from Eq. (72) as products of the two-particle ones – see Fig. 3 for a schematic guide. In expressing the difference $p_\ell - p_j$ of the momenta, our choice of permutations $\pi$ and $\pi'$ is arguably the simplest to factorise. For more complicated permutations, the factorisation of the scattering matrix can be determined, for example, by applying the Steinhaus-Johnson-Trotter algorithm. The two-particle scattering matrix is computed in Appendix C and reads

$$S_{b_1,b_2}\binom{p_1, p_2}{p_2, p_1} = -1 + b_1(1 - b_2)\left(1 - e^{i(p_1 - p_2)}\right). \tag{76}$$

By plugging it in Eq. (75) we finally obtain

$$e^{i(\frac{L}{2} + M)(p_\ell - p_j)} = 1, \tag{77}$$

where $M$ is defined in Eq. (51). The constraints on the momenta given by Eqs. (74) and (77) are the Bethe Ansatz equations of the dual folded XXZ model. Once they are solved, the coefficients $Z_{p_1,\ldots,p_N}^{b_1,\ldots,b_N}$ are fixed up to normalization by Eq. (72). Using the factorized scattering matrix and Eq. (76), we find

$$\frac{Z_{p_1,\ldots,p_N}^{b_2,\ldots,b_N,b_1}}{Z_{p_1,\ldots,p_N}^{b_1,\ldots,b_N}} = (-1)^\eta e^{-i\frac{L}{2}p_\ell} \prod_{\substack{n=1 \\ n \neq \ell}}^{N} S_{b_n,b_{n+1}}\binom{p_n, p_\ell}{p_\ell, p_n} = (-1)^{N-1+\eta} e^{-i(\frac{L}{2}+M)p_\ell} e^{i\sum_{n=1}^{N} b_n(1-b_{n+1})p_n}, \tag{78}$$

where, by virtue of Eq. (77), $p_\ell$ can be any of the momenta satisfying the Bethe equations.

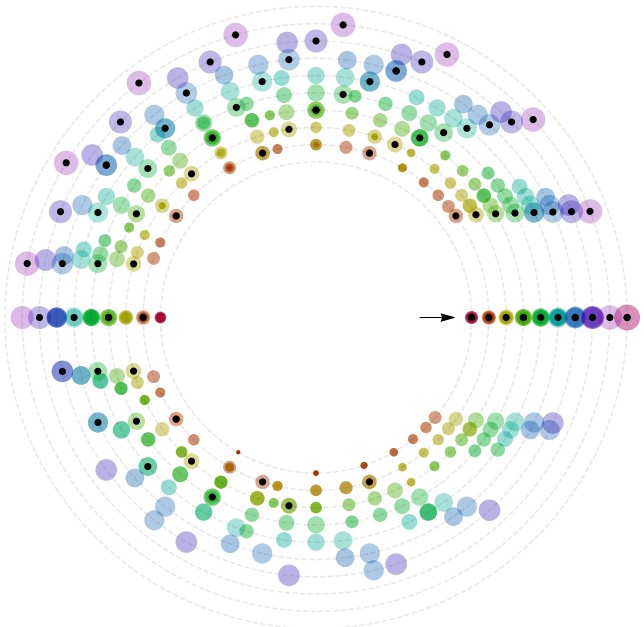

Figure 4: The range of the quantum numbers of the eigenstates of $\tilde{\mathbf{H}}_F^0$ for all possible configurations in a chain with $L = 18$ spins – *cf.* Eq. (90). The angular variable is $2\pi g I_0/N + \pi(g-1)$ mod $2\pi$; the radius is proportional to the maximal value of $I_{\ell>1}$ that can be reached for given $I_0$ (each dashed circle corresponds to an integer from $L/2$ to $L$). The arrow points at the circle representing the noninteracting sector. Symbols become larger and more transparent as the number of particles increases, whereas the colour represents the maximal $I_{\ell>1}$ associated with $I_0 = 0$, for each given configuration. The black dots identify points that are associated *also* with jammed configurations – the only ones in which, independently of $I_0$, the maximal value of $I_{\ell>0}$ is $\frac{L}{2} + M - 1$.

In summary, any Bethe state is specified by the configuration, $\underline{b} = (b_1, \ldots, b_N)$, of $N$ particles (equivalently, the set $B_N^{(e)}$ of its distinct cyclic permutations) and the set of $N$ solutions $\{p_1, \ldots, p_N\}$ to the Bethe equations. Its energy is the sum of the single-particle eigenvalues, which are computed in Appendix C. Specifically, we have

$$E_{B_N^{(e)}}(p_1, \ldots, p_N) = 4J \sum_{\ell=1}^{N} \cos p_\ell. \tag{79}$$

### 5.1.1   Quantum numbers

Remarkably, Bethe equations (74) and (77) can be solved explicitly, allowing us to overcome one of the complications of the Bethe Ansatz description. Specifically, the momenta $p_1, \ldots, p_N$ are given by

$$p_\ell = \frac{2\pi}{\frac{L}{2}+M}\left(I_\ell + \frac{2M}{LN}\sum_{j=1}^{N} I_j\right) + \frac{2\pi\eta}{L} + \frac{4\pi}{LN}I_0, \tag{80}$$

where $I_\ell \in \mathbb{Z}$ and $I_0 \in \{0, 1, \dots, N-1\}$. The total momentum $P = \sum_{\ell=1}^N p_\ell$ in turn satisfies

$$P = \frac{4\pi}{L} \sum_{j=0}^{N} I_j + \frac{2\pi N \eta}{L}. \tag{81}$$

Importantly, the phases $e^{ip_\ell}$, $\ell = 1, \dots, N$, do not change if one shifts $I_0 \to I_0 - M$ and one of the other quantum numbers, $I_n \to I_n + \left(\frac{L}{2} + M\right)$. Hence, all independent solutions can be found by imposing the folllowing constraints

$$0 \le I_1 < I_2 < \dots < I_N < \frac{L}{2} + M. \tag{82}$$

In practice, there are sets of quantum numbers corresponding to zero-norm states. This happens because there are other conditions, hidden in Eq. (78), that relate the coefficients of the state. To see that, let us introduce the cardinality $|B_N^{(e)}|$ of the set $B_N^{(e)} = \{\underline{b}\}_c$, i.e., the number of independent (distinct) configurations, equivalent to each other up to cyclic permutations. In the absence of any particular symmetry $|B_N^{(e)}| = N$. More generally, we have

$$|B_N^{(e)}| = \min\{m \mid b_{m+n} = b_n, \forall n\}. \tag{83}$$

For example, configuration $\underline{b} = (1, 0, 1, 1, 0, 1, 1, 0, 1, 1, 0, 1)$, for $N = 12$, has a *unit cell* $(1, 0, 1)$ that repeats itself. Its length is the cardinality $|B_N^{(e)}| = 3$. Since the collection of such unit cells forms the entire sequence $\underline{b}$, the cardinality $|B_N^{(e)}|$ is necessarily a divisor of $N$. Thus, we can define the quotient

$$g := \frac{N}{|B_N^{(e)}|} \in \mathbb{Z}_>, \tag{84}$$

which is clearly also a divisor of the number $M$ of oriented domain walls $(1, 0)$ in the configuration $\underline{b}$, i.e., $M/g \in \mathbb{Z}_>$. In order to have an eigenstate with a nonzero norm, condition (73) must be replaced by

$$\prod_{\underline{b} \in B_N^{(e)}} \frac{Z_{p_1, \dots, p_N}^{b_2, \dots, b_N, b_1}}{Z_{p_1, \dots, p_N}^{b_1, \dots, b_N}} = 1, \tag{85}$$

i.e., the product on its left-hand-side needs to be truncated to avoid repeated sequences $\underline{b}$. For an arbitrary subset of $N/g$ momenta $\{p_{\ell_j}\}$, $j = 1, \dots, N/g$, we can now rewrite this condition using Eq. (78), obtaining a new Bethe equation for the total momentum

$$\boxed{e^{i\left(\frac{L}{2} + M\right) \sum_{j=1}^{N/g} p_{\ell_j}} = (-1)^{(\eta + g - 1)\frac{N}{g}} e^{i\frac{M}{g} P}.} \tag{86}$$

Together with the constraint (77) imposed on the differences of momenta, this equation is finally solved by

$$p_\ell = \frac{2\pi}{\frac{L}{2} + M} \left(I_\ell + \frac{2M}{LN} \sum_{j=1}^N I_j\right) + \frac{2\pi}{L}(\eta + g - 1) + \frac{4\pi g}{NL} I_0, \tag{87}$$

where

$$0 \le I_0 < \frac{N}{g} \qquad \text{and} \qquad 0 \le I_1 < I_2 < \dots < I_N < \frac{L}{2} + M. \tag{88}$$

Despite the fact that these conditions are sufficient to generate a complete basis of eigenstates with nonzero norm, the same rapidities (modulo $2\pi$) can in fact be generated by different sets of quantum numbers. For example, for $L = 6$ there are 85 different configurations of quantum numbers satisfying Eq. (88) but only 64 independent states; for $L = 4$ there are 21 different configurations but only 16 independent states.

We conjecture that the independent eigenstates can be obtained by exhausting all choices of the quantum numbers that satisfy the following constraints:

$$0 \leq I_0 \leq I_0^{\max}, \qquad 0 \leq I_1 < I_2 < \ldots < I_N < F[I_0], \tag{89}$$

where the functional $F[I_0]$ and $I_0^{\max}$ are such that

$$\sum_{I_0=0}^{I_0^{\max}} \binom{F[I_0]}{N} = d_{B_N^{(e)}}, \qquad \frac{L}{2} + M \geq F[j] \geq F[j+1], \qquad I_0^{\max} < \frac{N}{g}. \tag{90}$$

Here, $d_{B_N^{(e)}}$ is the size of the space corresponding to configuration $B_N^{(e)}$. It can be computed by counting all the product states associated with the configuration. To that aim, let us call $2\{\ell_1', \cdots, \ell_{N_+}'\}$ the positions of spins up on even sites and $n_j$ the number of spins up on odd sites between $2\ell_j'$ and $2\ell_{j+1}'$ ($N_+ + \sum_{j=1}^{N_+} n_j = N$). Then we have

$$d_{B_N^{(e)}} = \sum_{\pi_c} \sum_{\ell_1' < \ell_2' < \cdots < \ell_{N_+}'}^{L/2} \binom{\ell_1' + \frac{L}{2} - \ell_{N_+}'}{n_{\pi_c(N_+)}} \prod_{j=1}^{N_+ - 1} \binom{\ell_{j+1}' - \ell_j'}{n_{\pi_c(j)}}, \tag{91}$$

where each binomial counts, in how many ways $n_j$ spins up can be distributed among the odd sites between $2\ell_j'$ and $2\ell_{j+1}'$. The first sum is over all the cyclic permutations that bring the sequence $(n_1, \ldots, n_{N_+})$ into an independent one, and thus exhausts all possible distributions of spins up on the odd sites. Our numerical investigations based on chains up to $L = 20$ sites suggest that, whenever possible, the bounds $F[I_0]$ are such that $I_0^{\max} = \frac{N}{g} - 1$. The only circumstance where we found the theoretical maximal value $\frac{N}{g} - 1$ unreached is when $\frac{L}{2} + M = N$, namely when the eigenstate is jammed – see Section 4.2. In that case we have $F[I_0] = \frac{L}{2} + M$ and $I_0^{\max} = \frac{L}{2g} - 1$. For $L = 18$, all possible sets of quantum numbers are depicted in Figure 4.

Finally, we note that the quantum number $I_0$ can be incorporated into the following *rational quantum numbers*

$$J_\ell = I_\ell + \frac{g}{N} I_0 + \frac{\eta + g - 1}{2}, \tag{92}$$

which live in the affine lattice $\mathbb{Z} + g I_0/N + (\eta + g - 1)/N$. In terms of them we have

$$p_\ell = \frac{2\pi}{\frac{L}{2} + M} \left( J_\ell + \frac{2M}{NL} \sum_{j=1}^N J_j \right), \qquad P = \frac{4\pi}{L} \sum_{j=1}^N J_j, \tag{93}$$

whence the one-to-one correspondence between the $N$ momenta and the $N$ quantum numbers becomes apparent. Rational quantum numbers, however, depend on the state. In this regard, the correspondence between $\{p_\ell\}$ and $\{J_\ell\}$ somewhat differs from the standard scenario that arises in models solvable via standard Bethe Ansatz techniques.

### 5.1.2  Degeneracy of momenta

A fixed $N$-tuple of momenta $\{p_\ell\}$ that solve Eqs. (77) and (86) describes multiple independent states, corresponding to distinct sets $B_N^{(e)}$. First of all, we note that $M$ is fixed by the set of momenta $\{p_\ell\}$. This can be proved by reductio ad absurdum. Let us indeed assume that the same set of momenta solves the Bethe Ansatz equations both with $M$ and with some $M' > M$:

$$e^{i(\frac{L}{2}+M)(p_\ell-p_j)} = e^{i(\frac{L}{2}+M')(p_\ell-p_j)} = 1. \tag{94}$$

The ratio of the left- and the right-hand side of the first equality yields $e^{i(M-M')(p_\ell-p_1)} = 1$. Since $M' - M \le N/2$, we can not have more than $N/2$ independent solutions to the latter equation in the interval $(0, 2\pi)$. On the other hand, the set $\{p_\ell\}$ consists of $N$ independent momenta, so there can not be any solution to the Bethe equations with $M' > M$ and the same set of momenta.

At fixed $g$ and $\{p_\ell\}$, the degeneracy $d_{g,\{p_\ell\}}$ is obtained by counting the configurations $\underline{b} = (b_1, \ldots, b_N)$ (with the value of $M$ fixed by the momenta) that, *up to cyclic permutations*, yield the same $g$. It depends on the size $N/g$ of the unit cell in the periodic structure of $\underline{b}$, and on the number $M/g$ of subsequences $(1, 0)$ in this unit cell:

$$d_{g,\{p_\ell\}} \equiv d\left(\tfrac{N}{g}, \tfrac{M}{g}\right). \tag{95}$$

To compute the degeneracy, consider first all sequences $\underline{b}$ of ones and zeros, with $M$ subsequences $(1, 0)$ (including also $(b_N, b_1)$, if $(b_N, b_1) = (1, 0)$ – see Eq. (51)). The number of such sequences is

$$2 \sum_{N_1=M}^{N-M} \binom{N_1-1}{M-1}\binom{N-N_1}{M} = 2\binom{N}{2M}, \tag{96}$$

where each term in the sum on the left-hand side represents the number of binary sequences at fixed $M$, composed of $N_1$ ones and $N - N_1$ zeros, and ending with a zero, $b_N = 0$. The sum is over all $N_1$ compatible with the existence of $M$ subsequences $(1, 0)$, while the prefactor 2 comes from counting also the sequences that end with a one instead of a zero, *i.e.*, $b_N = 1$.

On the other hand, the number of $N$-digit binary sequences with a unit cell of size $N/g$ reads $\frac{N}{g}d(\frac{N}{g}, \frac{M}{g})$, and thus

$$\sum_{g|N,M} \frac{N}{g}d\left(\tfrac{N}{g}, \tfrac{M}{g}\right) = 2\binom{N}{2M}, \tag{97}$$

where the sum runs over all common divisors of $N$ and $M$, *i.e.*, over all possible unit cells. Equality (97) holds also, if we substitute $N \to N/g'$ and $M \to M/g'$, for any common divisor $g'$ of $N$ and $M$. Let us now define square matrices $D, G$, and vectors $\vec{x}, \vec{y}$, whose dimensions are equal to the number of common divisors of $N$ and $M$, as follows

$$D_{m,n} = \begin{cases} 1, & m \mid n, \\ 0, & \text{otherwise}, \end{cases} \quad G_{m,n} = n\,\delta_{m,n}, \quad x_n = d\left(\tfrac{N}{n}, \tfrac{M}{n}\right), \quad \text{and} \quad y_n = \frac{2}{N}\binom{N/n}{2M/n}. \tag{98}$$

Collecting all the equations of the form (97) together, we find a matrix equation $D \cdot G^{-1} \cdot \vec{x} = \vec{y}$, which can be inverted to obtain the degeneracies $\vec{x} = G \cdot D^{-1} \cdot \vec{y}$. The inverse matrix $D^{-1}$ can be obtained from the Möbius inversion formula, which gives

$$D_{m,n} \equiv \sum_{k|n} \delta_{m,k} \quad \implies \quad \delta_{m,n} = \sum_{k|n} \mu\left(\tfrac{n}{k}\right)D_{m,k} = \sum_{k\in\mathbb{Z}_>} D_{m,k}\left[\mu\left(\tfrac{n}{k}\right)D_{k,n}\right], \tag{99}$$

for any two positive integers $n, m \in \mathbb{Z}_>$. Here, $\mu(n)$ denotes the Möbius function, which yields the sum of all primitive $n$-th roots of unity. If $n$ and $m$ are common divisors of $N$ and $M$, we can restrict the sums on the right-hand side of Eq. (99) to $k \mid N, M$, whence we recognize the elements of the inverse matrix $D^{-1}$ as $(D^{-1})_{k,n} = \mu\left(\frac{n}{k}\right)D_{k,n}$ (all indices now run over the common divisors of $N$ and $M$). Finally, plugging this into $\vec{d} = G \cdot D^{-1} \cdot \vec{v}$, gives the degeneracies

$$
d_{g,\{p_\ell\}} = \frac{2g}{N} \sum_{k|N/g,M/g} \mu(k) \binom{N/(gk)}{2M/(gk)}. \tag{100}
$$

**Generic states**

In the thermodynamic limit $N, M, L \to \infty$ with the ratios $\xi = \frac{L}{2N}$ and $\mu = M/N$ fixed, the vast majority of states have $g = 1$, *i.e.*, the configurations that characterise them have no periodic substructure. This can be inferred from Eq. (100) in the following way. First, we remind the reader that the number of distinct sets of momenta associated with a configuration depends only on $N$, $M$, and $g$. In addition, such a number is maximal when $g = 1$; indeed, for a larger $g$ the norm of some states becomes zero. The total number of states with given $g$, $N$, and $M$ is then proportional to $d_{g,\{p_\ell\}}$, the proportionality factor being maximal for $g = 1$. Thus we have

$$
\frac{\#[\text{states with } N, M, \text{ and } g = 1]}{\#[\text{states with } N, M]} \geq \frac{d_{1,\{p_\ell\}}}{\sum_{g|N,M} d_{g,\{p_\ell\}}}. \tag{101}
$$

Using Eq. (100), we can easily bound $d_{1,\{p_\ell\}}$ from below. In particular we have

$$
d_{1,\{p_\ell\}} \geq \frac{2}{N}\binom{N}{2M} - \frac{2\sqrt{M}}{N}\binom{N/2}{M}, \tag{102}
$$

where we have isolated the first term and bounded the rest from below by replacing $k$ with 2 in each term (in particular, we have replaced $\mu(k) \to -1$). Finally, we have used the fact that the number of common divisors is smaller than $\sqrt{M}$. Analogously, we can bound $d_{g,\{p_\ell\}}$, for $g > 1$, from above as

$$
d_{g,\{p_\ell\}} \leq \frac{4\sqrt{M}}{N}\binom{N/2}{M}, \qquad g > 1, \tag{103}
$$

which comes from replacing $k$ with 1 (in particular $\mu(k) \to 1$) and $g$ with 2, using again that the number of common divisors is smaller than $\sqrt{M}$. Plugging Eqs. (102) and (103) into Eq. (101) yields

$$
\frac{\#[\text{states with } N, M, \text{ and } g = 1]}{\#[\text{states with } N, M]} \geq \frac{1}{1 + (\sqrt{M} - 1)\frac{\frac{4\sqrt{M}}{N}\binom{N/2}{M}}{\frac{2}{N}\binom{N}{2M} - \frac{2\sqrt{M}}{N}\binom{N/2}{M}}} \sim 1 - e^{-\gamma N} \tag{104}
$$

for some $\gamma > 0$ depending on $M$ and $N$. That is to say, the number of states with $g > 1$ is exponentially smaller than the number of states with $g = 1$. In view of this, we refer to the Bethe states with $g = 1$ as *generic states*.

**Degeneracy at fixed staggered magnetisation**

We consider here the effect of fixing the staggered magnetisation $S_-^z$, defined as the expectation value of the staggered spin along the $z$-axis – see Eq. (39). In a Bethe state at fixed

configuration, $S_-^z$ is the difference between the numbers $N_0$ and $N_1$ of particles $\oplus$ and $\ominus$, respectively: $S_-^z = N_0 - N_1$. The number of binary sequences $\underline{b}$ at fixed $N = N_0 + N_1$, $M$, and $S_-^z$ reads

$$\binom{N_0 - 1}{M - 1}\binom{N_1}{M} + \binom{N_1 - 1}{M - 1}\binom{N_0}{M} = \binom{\frac{N+S_-^z-2}{2}}{M - 1}\binom{\frac{N-S_-^z}{2}}{M} + \binom{\frac{N-S_-^z-2}{2}}{M - 1} - 1\binom{\frac{N+S_-^z}{2}}{M}, \tag{105}$$

where the first (second) term on the left-hand side counts the binary sequences of $N_0$ zeros and $N_1$ ones ending with one (zero). Note that, when the staggered magnetisation is not fixed, summing the left-hand side of Eq. (105) over the allowed values of $N_1$ yields exactly Eq. (96).

For large $N$, the degeneracy at fixed staggered magnetisation can be approximated by the leading-order contribution to Eq. (105) divided by the number $N$ of cyclic permutations of the binary sequence $\underline{b}$:

$$d_{g,S_-^z,\{p_\ell\}} \sim \frac{2}{N}\binom{\frac{N+S_-^z}{2}}{M}\binom{\frac{N-S_-^z}{2}}{M}. \tag{106}$$

## 5.2 Odd number of sites

By analogy with the even $L$ case, the eigenstates in the sector characterized by the configuration $B_N^{(o)} = \{(\underline{b}; 1 - \underline{b})\}_c$, where $\underline{b} = (b_1, \ldots, b_N)$ and $1 - \underline{b} = (1 - b_1, \ldots, 1 - b_N)$, can be represented by a set of $N$ momenta (rapidities) $\{p_1, \ldots, p_N\}$. We use the Bethe Ansatz

$$|p_1, \ldots, p_N\rangle_{B_N^{(o)}} =$$

$$\sum_{(\underline{b};1-\underline{b})\in B_N^{(o)}} \sum_{\ell'_1=1}^{\frac{L-1}{2}+b_1} \sum_{\ell'_2=\ell'_1+\lceil\frac{b_2-b_1+1}{2}\rceil}^{\frac{L-1}{2}+b_2} \cdots \sum_{\ell'_N=\ell'_{N-1}+\lceil\frac{b_N-b_{N-1}+1}{2}\rceil}^{\frac{L-1}{2}+b_N} c_{\ell'_1,\ldots,\ell'_N}^{b_1,\ldots,b_n}(p_1,\ldots,p_N)\prod_{j=1}^N \boldsymbol{\sigma}_{2\ell'_j-b_j}^x |\phi\rangle, \tag{107}$$

where the ceiling function in the lower bounds of the sums accounts for the possibility of $\ominus$ and $\oplus$ sharing the same macrosite, i.e., $\underline{\ominus\oplus}$. The coefficients are again given in Eq. (69), but now satisfy boundary conditions

$$c_{\ell'_1,\ldots,\ell'_N}^{b_1,\ldots,b_N}(p_1,\ldots,p_N) = (-1)^\eta c_{\ell'_2,\ldots,\ell'_n,\ell'_1+\frac{L+1}{2}-b_1}^{b_2,\ldots,b_N,1-b_1}(p_1,\ldots,p_N) \tag{108}$$

that follow from $\boldsymbol{\sigma}_{L+n}^x = (-1)^\eta \boldsymbol{\sigma}_n^z$ and account for the transmutation of the particle crossing the boundary. Applying the Ansatz to the boundary conditions and removing the dependence on $\ell'_j$ we then find

$$\frac{Z_{p_1,\ldots,p_N}^{b_1,\ldots,b_N}}{Z_{p_1,\ldots,p_N}^{b_2,\ldots,b_N,1-b_1}} =$$

$$(-1)^\eta S_{b_1,\ldots,b_N}\binom{p_{\pi(1)},\ldots,p_{\pi(N)}}{p_1,\ldots,p_N}S_{b_2,\ldots,b_N,1-b_1}\binom{p_1,\ldots,p_N}{p_{\pi(2)},\ldots,p_{\pi(N)},p_{\pi(1)}}e^{i(\frac{L+1}{2}-b_1)p_{\pi(1)}}. \tag{109}$$

As a result of the factorisation of the scattering matrices, shown in Fig. 3, and of the explicit form of the two-particle one, (76), this is equivalent to

$$\frac{Z_{p_1,\ldots,p_N}^{b_1,\ldots,b_N}}{Z_{p_1,\ldots,p_N}^{b_2,\ldots,b_N,1-b_1}} = (-1)^{N-1+\eta}e^{-i\sum_{k=1}^{N-1}b_k(1-b_{k+1})p_k-ib_Nb_1p_N}e^{i(\frac{L+1}{2}+M_-)p_j}. \tag{110}$$

Here $M_-$, defined as

$$M_- := \sum_{k=1}^{N-1} b_k(1 - b_{k+1}) - (1 - b_N)b_1, \tag{111}$$

is invariant under "anti-cyclic" permutations of the configuration: $b_{N+1} \equiv 1 - b_1$. Since the ratio (110) holds for any momentum $p_j$, we obtain

$$\boxed{e^{i\left(\frac{L+1}{2} + M_-\right)(p_\ell - p_j)} = 1.} \tag{112}$$

On the other hand, the analogue of Eq. (73) with $2N$ cyclic permutations instead of $N$ yields the conservation of momentum

$$\boxed{e^{iL \sum_{j=1}^{N} p_j} = 1.} \tag{113}$$

The energy is again given by Eq. (79), *i.e.*, $E_{B_N^{(o)}}(p_1, \ldots, p_N) = E_{B_N^{(e)}}(p_1, \ldots, p_N)$.

**Quantum numbers**

Like in the even case, the Bethe equations can be solved explicitly:

$$p_\ell = \frac{2\pi}{\frac{L+1}{2} + M_-}\left(I_\ell + \frac{1 + 2M_-}{NL}\sum_{j=1}^{N} I_j\right) + \frac{2\pi}{NL}I_0, \tag{114}$$

where $I_\ell \in \mathbb{Z}$ and $I_0 \in \{0, 1, \ldots, 2N-1\}$. The phases $e^{ip_\ell}$, for $\ell = 1, \ldots, N$, do not change under the transformation $I_0 \to I_0 - 1 - 2M_-$ and $I_n \to I_n + \left(\frac{L+1}{2} + M_-\right)$, for a given $n > 0$. Therefore, all the independent solutions can be found by imposing

$$0 \le I_1 < I_2 < \ldots < I_N < \frac{L+1}{2} + M_-. \tag{115}$$

To avoid the zero-norm Bethe states, Eq. (113) has to be amended similarly as in the case of even $L$. Specifically, if the configuration $(\underline{b}; 1 - \underline{b})$ has a periodic substructure with a unit cell of size $|B_N^{(o)}| = 2N/g$ (copying the unit cell $g$-times yields the configuration), the product of ratios in Eq. (73) has to be truncated accordingly:

$$\prod_{(\underline{b}; 1-\underline{b}) \in B_N^{(o)}} \frac{Z_{p_1, \ldots, p_N}^{b_1, \ldots, b_N}}{Z_{p_1, \ldots, p_N}^{b_2, \ldots, b_N, 1-b_1}} = 1. \tag{116}$$

Using the explicit form of the ratio, given in Eq. (110), this becomes

$$e^{i\left(\frac{L+1}{2} + M_-\right)\sum_{j=1}^{2N/g} p_{\ell_j}} = (-1)^{\frac{2N}{g}(N+\eta-1)} e^{i\left(\frac{1+2M_-}{g}\right)P}, \tag{117}$$

for any set of $2N/g$ momenta $\{p_{\ell_j}\}$, $j = 1, \ldots, 2N/g$. Together with Eq. (112), it is solved by

$$p_\ell = \frac{2\pi}{\frac{L+1}{2} + M_-}\left(I_\ell + \frac{1 + 2M_-}{NL}\sum_{j=1}^{N} I_j\right) + \frac{2\pi g}{NL}I_0 + \frac{2\pi}{L}(N + \eta - 1). \tag{118}$$

Remarkably, calculation of the momenta shows that they do not depend on $\eta$. In fact, it seems that the last term in Eq. (118) can safely be ignored.

We warn the reader that, like in the even case, a set of rapidities could be generated by different sets of quantum numbers and, in order to have a one-to-one correspondence between quantum numbers and momenta, the allowed values for the quantum numbers should be appropriately reduced.

We conclude here the discussion about Bethe states; for the sake of simplicity we assume, in the rest of the paper, that the number $L$ of spins is even.

## 5.3    A family of local conservation laws

The local conservation laws of integrable systems are usually constructed within the framework of algebraic Bethe Ansatz. Indeed, the range of a charge cannot be easily inferred from coordinate Bethe Ansatz. Nevertheless, we conjecture that the local conservation laws introduced in Section 3.1 have single-particle eigenvalues equal to

$$q_n^+(p) = 4J\cos(np), \qquad q_n^-(p) = 4J\sin(np). \tag{119}$$

They notably form a complete basis of square-integrable functions on a circle. We have numerically checked this conjecture up to $n = 4$ by comparing the spectrum of the local conservation laws obtained by imposing the vanishing of their commutator with $\mathbf{H}_F$ against the Bethe Ansatz predictions.[10]

Let us compute how many charges of this form are independent in a system of finite size $L$. To that aim, we introduce non-Hermitian charges $\mathbf{Z}_n = \frac{1}{4J}(\mathbf{Q}_n^+ + i\mathbf{Q}_n^-)$, with eigenvalues

$$\langle \mathbf{Z}_n \rangle = \sum_{\ell=1}^{N} e^{inp_\ell}. \tag{120}$$

The knowledge of the expectation value of a generic conservation law with eigenvalue $\sum_{\ell=1}^{N} f(p_\ell)$ is equivalent to the knowledge of the set of momenta $\{p_\ell\}$. Since the momenta are defined modulo $2\pi$, we can assume $f(p)$ to be a $2\pi$-periodic function and write it in the Fourier space

$$\sum_{\ell=1}^{N} f(p_\ell) = \sum_{\ell=1}^{N} \sum_{n\in\mathbb{Z}} \hat{f}_n e^{inp_\ell} = \sum_{\ell=1}^{N} \sum_{n\in\mathbb{Z}} \sum_{m=1}^{\frac{L}{2}+M} \hat{f}_{(\frac{L}{2}+M)n+m} e^{i2\pi n\left(\frac{g}{N}I_0 + \frac{\eta+g-1}{2} + \frac{M}{N}\frac{P}{2\pi}\right)} e^{imp_\ell} =$$
$$= \sum_{m=1}^{\frac{L}{2}+M} \langle \mathbf{Z}_m \rangle \sum_{n\in\mathbb{Z}} (-1)^{n(\eta+g-1)} \hat{f}_{(\frac{L}{2}+M)n+m} e^{i2\pi n\left(\frac{g}{N}I_0 + \frac{M}{N}\frac{P}{2\pi}\right)}. \tag{121}$$

This equation implies that the knowledge of $I_0$, $P$, and $\langle \mathbf{Z}_n \rangle$, with $n = 1, \ldots, \frac{L}{2} + M$, unambiguously fixes the momenta through a functional that does not depend on $L$ explicitly (the latter appears only in the range of the index of charges). In fact, also the dependence on $I_0$ and $P$ can be removed by specialising the identity to $f(p) = 1$:

$$N = \langle \mathbf{Z}_0 \rangle = (-1)^{\eta+g-1} \langle \mathbf{Z}_{\frac{L}{2}+M} \rangle \, e^{-i2\pi\left(\frac{g}{N}I_0 + \frac{M}{N}\frac{P}{2\pi}\right)}. \tag{122}$$

Using this, we can express Eq. (121) in terms of $\langle \mathbf{Z}_n \rangle$ and $\hat{f}_n$ only. Assuming, without loss of

---

[10]In the compressed basis the charges have reduced range. There, the comparison is possible and was done also for larger $n$.

generality, that the single-particle eigenvalue $f(p)$ is real, we thus obtain

$$N \langle \mathbf{Z}_{\frac{L}{2}+M-\ell} \rangle = \langle \mathbf{Z}_{\frac{L}{2}+M} \rangle \langle \mathbf{Z}_\ell \rangle^* \, . \tag{123}$$ [11]

This implies that, for any given $M$, only $\mathcal{O}\left(\frac{L}{2}+M\right)$ Hermitian charges are actually independent. Note also that this relation involves the eigenstates of the operator $\mathbf{M}$, which we have shown in section 5.1.2 to be completely determined by this family of charges (at finite $L$).

Since the number of independent conservation laws scales linearly with the system size, the Fourier decomposition in Eq. (121) suggests that, in the thermodynamic limit $N, M, L \to \infty$, at fixed $\xi = \frac{L}{2N}$ and $\mu = M/N$, the eigenvalue $\sum_{\ell=1}^N f(p_\ell)$ is linear in the expectation values $\langle \mathbf{Z}_n \rangle$, for $n \in \mathbb{Z}_> \equiv \mathbb{N}$. Indeed the last term of the expansion

$$\frac{1}{N} \sum_{\ell=1}^N f(p_\ell) = \hat{f}_0 + \frac{1}{N} \sum_{n=1}^{\frac{L}{2}+M} \left( \hat{f}_n \langle \mathbf{Z}_n \rangle + h.c. \right) + \frac{1}{N} \sum_{n=\frac{L}{2}+M+1}^\infty \left( \hat{f}_n \sum_{\ell=1}^N e^{inp_\ell} + h.c. \right), \tag{124}$$

vanishes in the thermodynamic limit, provided that the Fourier series of $f(p)$ converges.

Additional insight is gained by using Eq. (123) to express $\langle \mathbf{Z}_{\frac{L}{2}+M} \rangle$ in terms of $\langle \mathbf{Z}_{\frac{L}{2}} \rangle$ and $\langle \mathbf{Z}_M \rangle$, and similarly for charges with index $n > \frac{L}{2} + M$. In this way Eq. (121) can be shown to be equivalent to

$$\sum_{\ell=1}^N f(p_\ell) = \langle \mathbf{Z}_0 \rangle \hat{f}_0 + \sum_{\ell=1}^{\frac{L}{2}+M} \sum_{n \in \mathbb{Z}_\geq} \left( \langle \mathbf{Z}_\ell \rangle \hat{f}_{(\frac{L}{2}+M)n+\ell} \left[ \frac{\langle \mathbf{Z}_M \rangle}{\langle \mathbf{Z}_{\frac{L}{2}} \rangle^*} \right]^n + h.c. \right) =$$

$$= \langle \mathbf{Z}_0 \rangle \hat{f}_0 + \left( \sum_{\ell=1}^{\frac{L}{2}-1} \sum_{n \in \mathbb{Z}_\geq} \langle \mathbf{Z}_\ell \rangle \left[ \frac{\langle \mathbf{Z}_M \rangle}{\langle \mathbf{Z}_{\frac{L}{2}} \rangle^*} \right]^n \hat{f}_{(\frac{L}{2}+M)n+\ell} + \sum_{\ell=0}^M \sum_{n \in \mathbb{Z}_<} \langle \mathbf{Z}_\ell \rangle \left[ \frac{\langle \mathbf{Z}_M \rangle}{\langle \mathbf{Z}_{\frac{L}{2}} \rangle^*} \right]^n \hat{f}_{(\frac{L}{2}+M)n+\ell} + h.c. \right), \tag{125}$$

for finite $L, N$ and $M$. Since $M \leq \frac{L}{2}$ and $\langle \mathbf{Z}_0 \rangle = N$, this equation reveals that the momenta are completely determined by the expectation values of $L/2$ non-Hermitian operators or, equivalently, $L$ Hermitian ones. These operators extend the family of local charges to all possible ranges.

# 6 Low-energy limits

## Ground state

While the XXZ model with anisotropy $\Delta > 1$ is gapped antiferromagnetic, due to the absence of the rapidly oscillating part $\kappa^{-1} \mathbf{H}_I$, the folded Hamiltonian has a critical ground state. The latter is a Fermi sea with two chiral modes, *i.e.*, it is conformal critical, the central charge being $c = 1$. In view of this, we identify it with a stationary state in which the neighbouring momenta

---

[11] Note that this relation between expectation values can be lifted into a relation between operators (it holds for all their eigenstates). In particular, we have $\mathbf{\Pi}_M \left( \mathbf{S}^z + \frac{L}{2} \right) \mathbf{Z}_{\frac{L}{2}+M-\ell} = \mathbf{\Pi}_M \mathbf{Z}_{\frac{L}{2}+M} \mathbf{Z}_\ell^*$, where $\mathbf{\Pi}_M$ is the projector on the eigenspace of $\mathbf{M}$ with eigenvalue M.

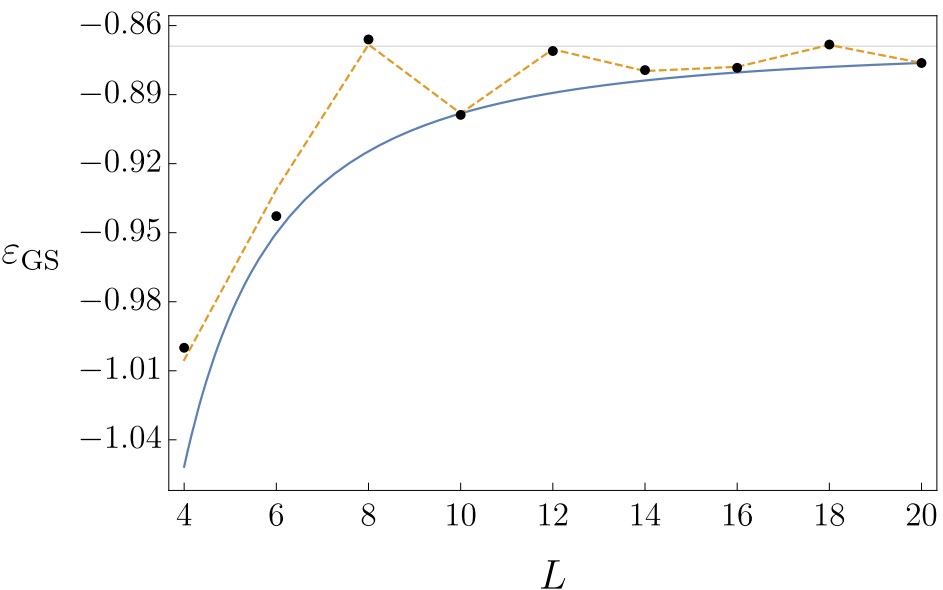

Figure 5: The ground state energy per unit length $\varepsilon_{\text{GS}} = E_{\text{GS}}/L$ as a function of the chain's length $L$ (black dots). The dashed orange piecewise linear curve is prediction (132). The horizontal grey line is the energy density in the thermodynamic limit. The blue curve includes only the leading CFT correction (133).

are occupied, *i.e.*, quantum numbers $I_j$, for $j = 1, \ldots, N$, are consecutive: $I_j = I_1 + j - 1$[12]. Using this Ansatz, the ground state energy reads

$$E(M, I_0, I_1, g, \eta) = \frac{4J \sin\left(\frac{2\pi N}{L+2M}\right)}{\sin\left(\frac{2\pi}{L+2M}\right)} \cos\left(\frac{4\pi}{L}\left[I_1 + \frac{g}{N}I_0 - 1 + \frac{N+g+\eta}{2}\right]\right). \tag{126}$$

Since $g$, $I_0$, and $I_1$ appear only in the argument of the cosine, we can minimize the energy by maximizing the ratio of sines and minimizing the cosine. At fixed average distance $\xi = \frac{L}{2N}$ between the particles, the maximal value of the ratio of sines is reached at $\mu = M/N = 1/2$. At finite $L$ this implies that $N$ is even and $g = N/2$: the configuration of the ground state is $B_N = \{(0, 1, 0, 1, \ldots, 0, 1)\}_c$. The cosine, on the other hand, is minimal when its argument is an odd multiple of $\pi$. Note that there are so many integer degrees of freedom that the minimum $-1$ is always reachable. Defining the variable[13]

$$\varphi = 2\pi - \frac{\pi}{\xi + \mu} \in [\pi, 2\pi), \tag{127}$$

which takes the value $\varphi = \frac{2\pi L}{N+L}$ in the ground state, the energy Ansatz reduces to

$$E(\varphi) = \frac{4J \sin \varphi}{\sin(L^{-1}\varphi)}. \tag{128}$$

---

[12]Whether these quantum numbers satisfy the conditions that guarantee the one-to-one correpondence between the quantum numbers and the momenta is irrelevant.

[13]The interval of allowed values of the variable $\varphi$ follows from the bound $\mu + \xi \geq 1$.

As a function of the parameter $\varphi$, the energy is minimal at the solution $\bar{\varphi}_L$ to the transcendental equation $\tan \bar{\varphi}_L = L \tan(L^{-1} \bar{\varphi}_L)$. The dependence of the parameter $\bar{\varphi}_L$ on the system size $L$ can be taken into account perturbatively as

$$\bar{\varphi}_L = \bar{\varphi}\left(1 + \frac{1}{3L^2}\right) + \mathcal{O}(L^{-4}), \tag{129}$$

where $\bar{\varphi}$ is the solution to $\tan \bar{\varphi} = \bar{\varphi}$ in the interval $[\pi, 2\pi)$, which is independent of the system size $L$. Expansion of the energy in $\Delta \varphi = \varphi - \bar{\varphi}_L$ now yields

$$E(\varphi) = 4JL\cos\bar{\varphi} + \frac{2J}{3L}\left(\sin\bar{\varphi}\tan\bar{\varphi} - 3L^2 \Delta\varphi^2 \cos\bar{\varphi}\right) + \mathcal{O}\left(L^{-2}, \Delta\varphi^3, \Delta\varphi^2 L^{-1}\right). \tag{130}$$

The final step is to minimise the lattice displacement $\Delta\varphi$, which is nonzero because $\frac{L+N}{2} = \frac{\pi L}{\varphi}$ is an integer. We find

$$L\Delta\varphi = \frac{\tan^2 \bar{\varphi}}{\pi}\lfloor\frac{\pi L}{\tan\bar{\varphi}}\rfloor - L\tan\bar{\varphi} + \mathcal{O}(L^{-1}), \tag{131}$$

which, plugged into Eq. (130), yields

$$\boxed{E_{\mathrm{GS}} = 4JL\cos\bar{\varphi} + \frac{2J}{L}\sin\bar{\varphi}\tan\bar{\varphi}\left(\frac{1}{3} - \left\{\frac{\tan\bar{\varphi}}{\pi}\lfloor\frac{\pi L}{\tan\bar{\varphi}}\rfloor - L\right\}^2\right) + o(L^{-1}),} \tag{132}$$

where $o(L^{-1})$ means that the correction approaches zero faster than $L^{-1}$ – see Fig. 5. The term originating in the displacement $\Delta\varphi$ is a clear lattice effect: it arises because $L$ and $N$ are (even) integers. On the other hand, the $\mathcal{O}(L^{-1})$ correction that remains when the integer condition is relaxed (i.e., keeping only the first term in the rounded brackets in Eq. (132)) is expected to be universal and describable by the underlying conformal field theory. Specifically, the prediction for periodic boundary conditions is [36, 37]

$$E_{\mathrm{GS}} \sim \frac{L}{2}\varepsilon_0 - \frac{\pi c}{3L}v_{\mathrm{F}}, \tag{133}$$

where $c$ is the central charge, $v_{\mathrm{F}}$ is the Fermi velocity, and $\varepsilon_0$ is the ground state energy density in the infinite volume, which is not universal (note that the volume is $L/2$ because the macrosites consist of two adjacent spins). Since the theory under investigation has two chiral modes, the central change $c$ is equal to 1. By equating the volume correction that we computed with the conformal field theory prediction we can then infer the Fermi velocity

$$v_{\mathrm{F}} = -\frac{2J\sin\bar{\varphi}\tan\bar{\varphi}}{\pi} \approx 2.7922813J. \tag{134}$$

This result is in agreement with the thermodynamic Bethe Ansatz calculation worked out in the second part of this work [35], as well as with the calculation in the constrained model, described by the Hamiltonian (32) – see Ref. [20]. This is another indication that constrained Hamiltonians, in which particles have a finite radius, correspond to low-energy limits of the asymptotically folded Hamiltonians.

Figure 5 compares the CFT correction against the full leading one (132) and the exact ground-state energy for small chains up to $L = 20$.

Finally, we note that the ground state has a finite negative magnetisation $\frac{1}{L}\langle \mathbf{S}^z\rangle = \frac{N}{L} - \frac{1}{2} = \frac{2\pi}{\varphi} - \frac{3}{2} \sim \frac{2\pi}{\bar{\varphi}} - \frac{3}{2} \approx -0.1$.

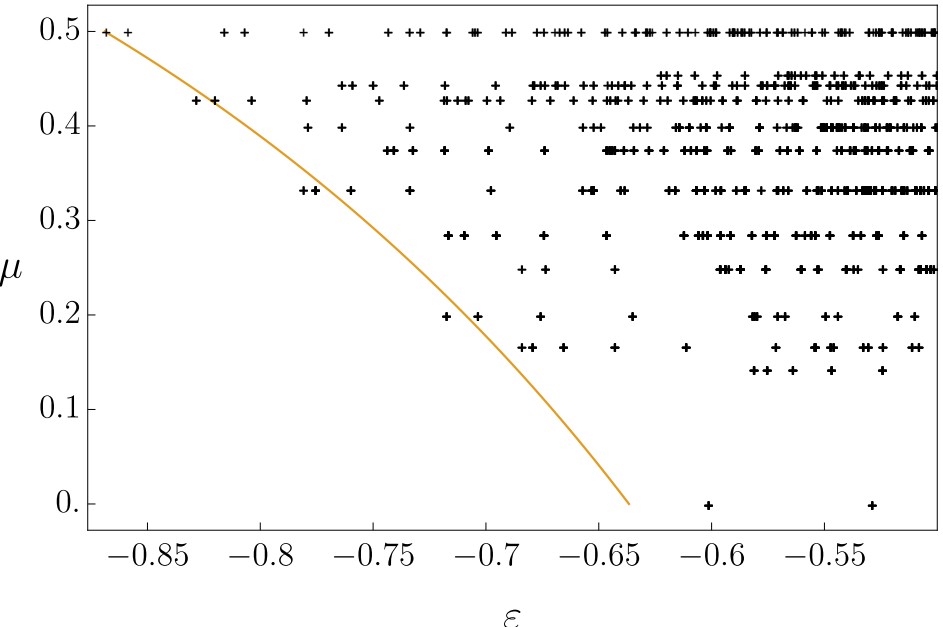

Figure 6: Energy per unit length $\varepsilon = E/L$ and $\mu = \frac{M}{N}$ of the eigenstates (black symbols) in the low-energy part of the spectrum of $\tilde{\mathbf{H}}_F^0$, for $L = 18$. The orange curve is prediction (145), which is valid in the thermodynamic limit $L \to \infty$.

## Average size of domains in minimal-energy states

In the previous subsection we have shown that, in the ground state, $M = N/2$ and hence $g = N/2$. The Ansatz (126) for the energy is, however, still valid if we impose a different value for $\mu$ and minimise the energy. For $\mu \neq 0$, the result will then be the minimal energy at which the groups of consecutive particles of the same species, *i.e.*, the domains, have an average size of $(2\mu)^{-1}$. As we will see, the minimal energy increases with the average number of consecutive identical particles. The exact calculation is very similar to the one carried out in the previous subsection. Again, the cosine in the Ansatz (126) can be replaced by $-1$, so that the energy reads

$$E(\varphi) = \frac{4J \sin \varphi}{\sin(L^{-1}[2\mu\,\varphi + (1 - 2\mu)\,2\pi])}, \tag{135}$$

where $\varphi$ is again given in Eq. (127) and depends on the parameter $\mu$. The energy is minimal at $\varphi = \bar{\varphi}_L$, satisfying the transcendental equation $\tan \bar{\varphi}_L = \frac{L}{2\mu} \tan \left(L^{-1}[2\mu\,\bar{\varphi}_L + (1 - 2\mu)2\pi]\right)$. As before, we take into account the dependence of the parameter $\bar{\varphi}_L$ on $L$ perturbatively,

$$\bar{\varphi}_L = \bar{\varphi}\left(1 + \frac{1}{3}\left[\frac{2\mu}{L}\right]^2\right) + \frac{4\pi\mu[1 - 2\mu]}{3L^2} + \mathcal{O}(L^{-4}), \tag{136}$$

where $\bar{\varphi}$ is the solution to $2\mu \tan \bar{\varphi} = 2\mu\,\bar{\varphi} + (1 - 2\mu)2\pi$. It lies in the interval $[\pi, 3\pi/2)$ and is independent of the system size $L$. We can now expand in $\Delta\varphi = \varphi - \bar{\varphi}_L$:

$$E(\varphi) = \frac{2JL}{\mu}\cos\bar{\varphi} + \frac{4J\mu}{3L}\left(\sin\bar{\varphi}\tan\bar{\varphi} - \frac{3\cos\bar{\varphi}}{4\mu^2}L^2\Delta\varphi^2\right) + \mathcal{O}\left(L^{-2}, \Delta\varphi^3, \Delta\varphi^2 L^{-1}\right). \tag{137}$$

The non-zero lattice displacement $\Delta\varphi$ is now obtained by imposing $\frac{\pi L}{2\mu\varphi + (1-2\mu)2\pi} = \frac{L}{2} + M \in \mathbb{N}$, which results in

$$L\Delta\varphi = \frac{2\mu\tan^2\bar{\varphi}}{\pi}\lfloor\frac{\pi L}{2\mu\tan\bar{\varphi}}\rfloor - L\tan\bar{\varphi} + \mathcal{O}(L^{-1}). \tag{138}$$

Finally, the energy is expressed as

$$E_{\min} = \frac{2JL}{\mu}\cos\bar{\varphi} + \frac{4J\mu}{L}\sin\bar{\varphi}\tan\bar{\varphi}\left(\frac{1}{3} - \left\{\frac{\tan\bar{\varphi}}{\pi}\lfloor\frac{\pi L}{2\mu\tan\bar{\varphi}}\rfloor - \frac{L}{2\mu}\right\}^2\right) + o(L^{-2}), \tag{139}$$

and using the conformal field theory prediction (133) we now find

$$v_{\mathrm{F}}(\mu) = -\frac{4J\mu}{\pi}\sin\bar{\varphi}\tan\bar{\varphi}. \tag{140}$$

This again agrees with the thermodynamic Bethe Ansatz calculation worked out in the second part of this work [35]. Differently from the genuine ground state with energy (132), the minimal energy states with energy $E_{\min}$ have exponentially large degeneracy; indeed only $\mu = 1/2$ (in the genuine ground state) fixes the configuration unambiguously.

A question that now arises is, how the ratio $\mu_{\min} = M/N$ changes as a function of the energy density

$$\varepsilon := \lim_{L\to\infty}\frac{E_{\min}}{L} = \frac{2J}{\mu_{\min}}\cos\bar{\varphi} + \mathcal{O}(L^{-2}) \tag{141}$$

in the minimal energy state. Here, $\bar{\varphi}$ depends on $\mu_{\min}$, and the dependence of the latter on $\varepsilon$ is what we wish to obtain. Eq. (141), on the one hand, implies

$$\tan\bar{\varphi} = \sqrt{\left(\frac{2J}{\varepsilon\mu_{\min}}\right)^2 - 1} + \mathcal{O}(L^{-2}), \tag{142}$$

where the positive branch of the square root has been chosen because the relevant solution $\bar{\varphi}$ to the transcendental equation $2\mu\tan\bar{\varphi} = 2\mu\bar{\varphi} + (1-2\mu)2\pi$ falls into the interval $[\pi, 3\pi/2)$. On the other hand, the transcendental equation, along with Eq. (141), also yields

$$\tan\bar{\varphi} = -\arccos\left(\frac{\varepsilon\mu_{\min}}{2J}\right) + \frac{\pi}{\mu_{\min}} + \mathcal{O}(L^{-2}), \tag{143}$$

where, again, the branch of arccos is chosen in such a way that $\bar{\varphi} \in [\pi, 3\pi/2)$. Noting that $\varepsilon < 0$, we now have

$$\frac{\varepsilon\mu_{\min}}{2J}\arccos\left(\frac{\varepsilon\mu_{\min}}{2J}\right) - \sqrt{1 - \left(\frac{\varepsilon\mu_{\min}}{2J}\right)^2} = \frac{\pi\varepsilon}{2J} + \mathcal{O}(L^{-2}). \tag{144}$$

We now recognise the left-hand side to be equal to $\int_1^{\frac{\varepsilon\mu_{\min}}{2J}} \mathrm{d}y\,\arccos y$; the solution can then be expressed implicitly as

$$\boxed{\mu_{\min}(\varepsilon) = \theta\left(-\frac{2J}{\pi} - \varepsilon\right)\frac{2J}{\varepsilon}f\left(\frac{\pi\varepsilon}{2J}\right) + \mathcal{O}(L^{-2}),} \tag{145}$$

where $f(x)$ is a solution to $\int_1^{f(x)}\mathrm{d}y\,\arccos y = x$ – see Fig. 6. The Heaviside step function $\theta$ comes from the fact that $\frac{\varepsilon\mu_{\min}}{2J}$ and $\frac{\pi\varepsilon}{2J}$ should both have the sign of $\varepsilon$, and Eq. (144) is consistent with such a constraint only if $\varepsilon \leq -2J/\pi$.

For the sake of completeness we also report the value of the minimal number $M$ of oriented domain walls in the eigenstates with a given energy density $\varepsilon$:

$$M_{\min}(\varepsilon) = \lfloor \frac{-\pi \varepsilon L}{4J\sqrt{1 - [f(\frac{\pi \varepsilon}{2J})]^2}} \rfloor - \frac{L}{2} + \mathcal{O}(1). \tag{146}$$

# 7  Conclusion

We have proposed a framework for studying time evolution in quantum many-body systems on intermediate time scales at large coupling constant. As an example, we have investigated the effective Hamiltonian that arises in the limit of large anisotropy $\Delta$ in the Heisenberg spin-1/2 XXZ model. We have developed a coordinate Bethe Ansatz that manifests the symmetries emerging in the strong coupling limit. Differently from the Bethe Ansatz solution of the XXZ model [38], the new Ansatz is based on two species of particles that live on macrosites, comprising two neighbouring spins. As a consequence, the Bethe states explicitly break the translational symmetry of the effective Hamiltonian. The Bethe states with both species of particles form an interacting sector, while those with a single species of particles form the noninteracting one. The latter sector can be mapped into an effective XX model, which is well known for its non-abelian set of conservation laws [15]. How they fit into the model's overall integrable structure is an interesting problem, left to future investigations. We have also identified excited states in which particles populate the lattice in a chaotic way and are packed so closely that they keep fixed positions. These jammed states form an exponentially large subspace of the Hilbert space, which we showed to span the eigenspace of the ground state of a quasilocal charge $\mathbf{M} - \mathbf{S}^z$. We also found that a subset of such states is stable under the $\mathcal{O}(\Delta^{-1})$ correction to the leading-order effective Hamiltonian. Whether or not the leading correction preserves integrability is still an open question. All the above properties make the proposed effective model a particularly convenient playground for studying pre-relaxation phenomena.

We have found that the ground-state energy and Fermi velocity agree with those calculated by Alcaraz and Bariev in the constrained XX Hamiltonian, describing finite-radius particles [20]. Their projected Hamiltonian corresponds to the four-vertex model and we speculate that it describes the low-energy sector of our effective Hamiltonian. In fact, Alcaraz and Bariev provided also a Bethe Ansatz for a range of larger particle radii; it would be interesting to see how the other constrained Hamiltonians fit in the Bethe Ansatz that we considered.

Finally, we remark that the simplicity of the Bethe Ansatz equations makes the effective Hamiltonian originating in the large-anisotropy limit of the XXZ model an excellent candidate for advancing the theory of generalised hydrodynamics in interacting systems. To set the stage for it, the second part of this work [35] will develop a thermodynamic Bethe Ansatz and a generalised hydrodynamic theory on the Euler scale.

# Acknowledgements

**Funding information.**  We thank Fabian Essler and Leonardo Mazza for useful discussions. This work was supported by the European Research Council under the Starting Grant No. 805252 LoCoMacro.

# A  Asymptotic expansion

For the sake of completeness, we recover the result of Ref. [16] and its generalisation by showing the asymptotic validity of decomposition (5), which is equivalent to

$$e^{-i\mathbf{H}(\kappa)t} = e^{-i\kappa^{-1}\mathbf{H}_I t}e^{-i\kappa\mathbf{B}_{z_t}(\kappa)}e^{-i\mathbf{H}_F(\kappa)t}e^{i\kappa\mathbf{B}_1(\kappa)}, \tag{147}$$

where we have used Eqs. (2) and the fact that $\mathbf{B}_z(\kappa)$ is a functional of the form

$$\mathbf{B}_z(\kappa) = \mathbf{B}\big[\{z^m\mathbf{F}_m\}_{m=1}^q, \{z^{-m}\mathbf{F}_m^\dagger\}_{m=1}^q, \mathbf{H}_F; \kappa\big]. \tag{148}$$

To show the asymptotic validity of Eq. (147), we now define the operator

$$\mathbf{V}(t) = e^{i(\mathbf{H}_F+\kappa^{-1}\mathbf{H}_I)t}e^{-i\mathbf{H}(\kappa)t}, \tag{149}$$

satisfying the differential equation

$$i\partial_t\mathbf{V}(t) = \sum_{m=1}^q e^{i\mathbf{H}_F t}[z_t^m\mathbf{F}_{m,t} + z_t^{-m}\mathbf{F}_{m,t}^\dagger]e^{-i\mathbf{H}_F t}\mathbf{V}(t), \tag{150}$$

where $z_t = e^{iJt/\kappa}$. Assuming decomposition (147), on the other hand, we have

$$\mathbf{V}(t) \overset{(147)}{=} e^{i\mathbf{H}_F t}e^{-i\kappa\mathbf{B}_{z_t}(\kappa)}e^{-i\mathbf{H}_F(\kappa)t}e^{i\kappa\mathbf{B}_1(\kappa)} \tag{151}$$

and hence

$$i\partial_t\mathbf{V}(t) \overset{(147)}{=} e^{i\mathbf{H}_F t}\Big(-\mathbf{H}_F + \big[-\kappa^{-1}Jz_t\partial_{z_t}e^{-i\kappa\mathbf{B}_{z_t}(\kappa)} + e^{-i\kappa\mathbf{B}_{z_t}(\kappa)}\mathbf{H}_F(\kappa)\big]e^{i\kappa\mathbf{B}_{z_t}(\kappa)}\Big)e^{-i\mathbf{H}_F t}\mathbf{V}(t). \tag{152}$$

If decomposition (147) is possible, this expression has to match Eq. (150), whence a gauge transformation

$$\kappa^{-1}Jz\partial_z + \mathbf{H}_F + \sum_{m=1}^q (z^m\mathbf{F}_m + z^{-m}\mathbf{F}_m^\dagger) = e^{-i\kappa\mathbf{B}_z(\kappa)}\Big[\kappa^{-1}Jz\partial_z + \mathbf{H}_F(\kappa)\Big]e^{i\kappa\mathbf{B}_z(\kappa)} \tag{153}$$

follows for any $z \in \mathbb{C}$, with $|z| = 1$. Here, $\partial_z$ denotes the derivation operator, *i.e.*, both sides of the equation should be read as if applied to some differentiable function of $z$.

The existence of a solution to this equation is the necessary and sufficient condition for the validity of Eq. (147). Using Eq. (4) and notation $\mathbf{A}_n^\dagger(\kappa) \equiv \mathbf{A}_{-n}(\kappa) \equiv \mathbf{A}_n^-(\kappa) \equiv \mathbf{A}_{-n}^+(\kappa)$ we can recast Eq. (153) as an infinite system of equations

$$(\ell + \delta_{\ell,0})\mathbf{A}_\ell(\kappa) = J^{-1}\delta_{\ell,0}\mathbf{H}_F - iJ^{-1}\theta(\ell \le q)\mathbf{F}_\ell + \kappa\Big[\mathbf{A}_\ell(\kappa), \mathbf{A}_0(k)\Big] +$$

$$+ [1 - (1-i)\delta_{\ell,0}]\sum_{n=1}^\infty \frac{(-i\kappa)^n}{(n+1)!}\sum_{\substack{\underline{m}\in\mathbb{Z}_>^{\times n}\\\underline{s}\in\{-1,1\}^{\times n}}}\Big[\mathbf{A}_{m_1}^{s_1}(\kappa), \big[\mathbf{A}_{m_2}^{s_2}(\kappa), \ldots$$

$$\ldots\Big[\mathbf{A}_{m_n}^{s_n}(\kappa), \Big(\Big\{\sum_{k=1}^n s_k m_k - \ell\Big\}\mathbf{A}_{\sum_{k=1}^n s_k m_k - \ell}^-(\kappa) + \kappa\Big[\mathbf{A}_{\sum_{k=1}^n s_k m_k - \ell}^-(\kappa), \mathbf{A}_0(\kappa)\Big]\Big)\Big]\ldots\Big]\Big], \tag{154}$$

where we have additionally defined $\mathbf{A}_0(\kappa) = J^{-1}\mathbf{H}_F(\{\mathbf{F}_m\}_{q=1}^m, \{\mathbf{F}_m^\dagger\}_{q=1}^m, \mathbf{H}_F; \kappa)$, while $\theta$ denotes the Heaviside step function. From this we can obtain the asymptotic expansion of $\mathbf{B}_z(\kappa)$, and hence of $\mathbf{A}_\ell(\kappa)$, about $\kappa = 0$. If we define $\mathbf{A}_\ell(\kappa) = \sum_{j=0}^\infty \mathbf{A}_{\ell,j}\kappa^j$, we finally find

$$
\mathbf{A}_{\ell,j} = \frac{\mathbf{H}_F \delta_{\ell,0} - i\theta(\ell \leq q)\mathbf{F}_\ell}{J(\ell + \delta_{\ell,0})}\delta_{j,0} + \frac{1}{\ell + \delta_{\ell,0}}\sum_{n=0}^{j-1}\Big[\mathbf{A}_{\ell,n}, \mathbf{A}_{0,j-n-1}\Big]+
$$

$$
+\frac{1 - (1-i)\delta_{\ell,0}}{\ell + \delta_{\ell,0}}\sum_{n=1}^{j}\frac{(-i)^n}{(n+1)!}\sum_{\substack{\underline{d}\in\mathbb{N}_0^{\times(n+1)} \\ \sum_{k=1}^{n+1}d_k=j-n}}\sum_{\substack{\underline{m}\in\mathbb{N}_1^{\times n} \\ \underline{s}\in\{-1,1\}^{\times n}}}\left(\sum_{k=1}^n s_k m_k - \ell\right)\Big[\mathbf{A}_{m_1,d_1}^{s_1}, \Big[\mathbf{A}_{m_2,d_2}^{s_2}, \cdots
$$

$$
\cdots\Big[\mathbf{A}_{m_n,d_n}^{s_n}, \mathbf{A}_{\sum_{k=1}^n s_k m_k-\ell,d_{n+1}}^{-}\Big]\cdots\Big]\Big]+ \tag{155}
$$

$$
+\frac{1 - (1-i)\delta_{\ell,0}}{\ell + \delta_{\ell,0}}\sum_{n=1}^{j-1}\frac{(-i)^n}{(n+1)!}\sum_{\substack{\underline{d}\in\mathbb{N}_0^{\times(n+2)} \\ \sum_{k=1}^{n+2}d_k=j-n-1}}\sum_{\substack{\underline{m}\in\mathbb{N}_1^{\times n} \\ \underline{s}\in\{-1,1\}^{\times n}}}\Big[\mathbf{A}_{m_1,d_1}^{s_1}, \Big[\mathbf{A}_{m_2,d_2}^{s_2}, \cdots
$$

$$
\cdots\Big[\mathbf{A}_{m_n,d_n}^{s_n}, \Big[\mathbf{A}_{\sum_{k=1}^n s_k m_k-\ell,d_{n+1}}^{-}, \mathbf{A}_{0,d_{n+2}}\Big]\Big]\cdots\Big]\Big].
$$

In particular, at the lowest orders in $\kappa$, for $q = 2$, we have

$$
\mathbf{H}_F(\kappa) = J\mathbf{A}_0(\kappa) \sim \mathbf{H}_F + \frac{\kappa}{J}\left([\mathbf{F}_1, \mathbf{F}_1^\dagger] + \frac{1}{2}[\mathbf{F}_2, \mathbf{F}_2^\dagger]\right) + \frac{\kappa^2}{J^2}\Big(\frac{1}{2}[\mathbf{F}_1, [\mathbf{F}_1, \mathbf{F}_2^\dagger]]+
$$

$$
+ \frac{1}{2}[\mathbf{F}_1^\dagger, [\mathbf{F}_1^\dagger, \mathbf{F}_2]] + i\frac{1}{2}[\mathbf{F}_1', \mathbf{F}_1^\dagger] + i\frac{1}{2}[\mathbf{F}_1^{\dagger\prime}, \mathbf{F}_1] + i\frac{1}{8}[\mathbf{F}_2', \mathbf{F}_2^\dagger] + i\frac{1}{8}[\mathbf{F}_2^{\dagger\prime}, \mathbf{F}_2]\Big),
$$

$$
\mathbf{A}_1(\kappa) \sim -i\frac{1}{J}\mathbf{F}_1 + i\frac{\kappa}{J^2}\left(-i\mathbf{F}_1' + \frac{3}{4}[\mathbf{F}_1^\dagger, \mathbf{F}_2]\right) + i\frac{\kappa^2}{J^3}\Big(\mathbf{F}_1'' + i\frac{1}{4}[\mathbf{F}_1^\dagger, \mathbf{F}_2]' - \frac{2}{3}[\mathbf{F}_1, [\mathbf{F}_1, \mathbf{F}_1^\dagger]] +
$$

$$
+ i\frac{5}{8}[\mathbf{F}_1^\dagger, \mathbf{F}_2'] - \frac{19}{48}[[\mathbf{F}_1, \mathbf{F}_2], \mathbf{F}_2^\dagger] + \frac{37}{48}[[\mathbf{F}_1, \mathbf{F}_2^\dagger], \mathbf{F}_2]\Big),
$$

$$
\mathbf{A}_2(\kappa) \sim -i\frac{1}{J}\mathbf{F}_2 + \frac{\kappa}{J^2}\frac{1}{4}\mathbf{F}_2' + i\frac{\kappa^2}{J^3}\Big(\frac{1}{8}\mathbf{F}_2'' + \frac{1}{24}[\mathbf{F}_1, [\mathbf{F}_1^\dagger, \mathbf{F}_2]] - \frac{1}{8}[\mathbf{F}_1^\dagger, [\mathbf{F}_1, \mathbf{F}_2]] - \tag{156}
$$

$$
- \frac{1}{12}[\mathbf{F}_2, [\mathbf{F}_2, \mathbf{F}_2^\dagger]] - i\frac{1}{4}[\mathbf{F}_1', \mathbf{F}_1]\Big),
$$

$$
\mathbf{A}_3(\kappa) \sim -i\frac{\kappa}{J^2}\frac{1}{12}[\mathbf{F}_1, \mathbf{F}_2] + i\frac{\kappa^2}{J^3}\left(-i\frac{7}{36}[\mathbf{F}_1, \mathbf{F}_2]' + i\frac{5}{24}[\mathbf{F}_1, \mathbf{F}_2'] + \frac{5}{48}[[\mathbf{F}_1^\dagger, \mathbf{F}_2], \mathbf{F}_2]\right),
$$

$$
\mathbf{A}_4(\kappa) \sim \frac{\kappa^2}{J^3}\frac{1}{32}[\mathbf{F}_2', \mathbf{F}_2],
$$

$$
\mathbf{A}_5(\kappa) \sim -i\frac{\kappa^2}{J^3}\frac{1}{240}[[\mathbf{F}_1, \mathbf{F}_2], \mathbf{F}_2],
$$

$$
\mathbf{A}_{\ell>5}(\kappa) \sim \mathcal{O}(\kappa^3),
$$

where $(\bullet)'$ stands for the commutator $i[\mathbf{H}_F, \bullet]$. The case $q = 1$ is obtained by setting $\mathbf{F}_2 = 0$ in the above equations.

# B Folded picture in noninteracting spin-$1/2$ chains

In this appending we consider an example – a noninteracting spin-1/2 chain with $\mathbf{H}_I \propto \sum_\ell \boldsymbol{\sigma}_\ell^z$ – in which the folded picture can be worked out in a non-perturbative way. Other cases could

be addressed by applying first a noninteracting mapping that sends $\mathbf{H}_I$ into an operator proportional to $\sum_\ell \boldsymbol{\sigma}_\ell^z$. Specifically, we consider Hamiltonians that can be written as

$$\mathbf{H} = \frac{1}{4} \sum_{\ell,n} \mathbf{a}_\ell \mathcal{H}_{\ell n} \mathbf{a}_n, \tag{157}$$

where $\mathbf{a}_\ell$ are the Majorana fermions defined in Eq. (21). The matrix $\mathcal{H}$ is conveniently written as a Fourier transform, which is used to define the so-called "symbol" of the noninteracting operator. Let us show the procedure by considering the Hamiltonian of the Ising model (18), with the matrix elements

$$\mathcal{H}_{2\ell-1+\imath,2n-1+\jmath} = \frac{iJ}{2}\big[\delta_{n,\ell+1}\sigma^- - \delta_{n,\ell-1}\sigma^+ + ih\delta_{\ell n}\sigma^y\big]_{\imath\jmath} = -\frac{J}{2}\int_{-\pi}^{\pi}\frac{\mathrm{d}k}{2\pi}e^{i(\ell-n)k}\sigma^y\big[h - e^{ik\sigma^z}\big]. \tag{158}$$

Its symbol is a $2 \times 2$ matrix function appearing in the Fourier transform

$$\hat{h}(e^{ik}) = -\frac{J}{2}\sigma^y\big[h - e^{ik\sigma^z}\big] = -\varepsilon(k)e^{-i\frac{1}{2}\theta(k)\sigma^z}\sigma^y e^{i\frac{1}{2}\theta(k)\sigma^z}, \tag{159}$$

where

$$e^{i\theta(k)} = \sqrt{\frac{1 - h^{-1}e^{ik}}{1 - h^{-1}e^{-ik}}}, \qquad \varepsilon(k) = \frac{J}{2}\sqrt{1 + h^2 - 2h\cos k}. \tag{160}$$

Analogously, the symbol of $\mathbf{H}_I$ is $-\frac{J}{2}\sigma^y$. A very important property is that the symbol of the commutator of two noninteracting operators is the commutator of the corresponding symbols. This can be readily exploited to exhibit a unitary transformation that maps $\mathbf{H}$ into an operator commuting with $\mathbf{H}_I$.

To that aim, let us consider the noninteracting operator $\mathbf{W}(h^{-1})$ represented by

$$\mathcal{W}_{2\ell-1+\imath,2n-1+\jmath}(h^{-1}) = -\frac{i}{4}\int_{-\pi}^{\pi}\frac{\mathrm{d}k}{2\pi}e^{i(\ell-n)k}\log\frac{1 - h^{-1}e^{ik}}{1 - h^{-1}e^{-ik}}\sigma_{\imath\jmath}^z. \tag{161}$$

From Eq. (159) and the Baker-Campbell-Hausdorf formula it follows that the noninteracting operator $e^{i\mathbf{W}(h^{-1})}\mathbf{H}e^{-i\mathbf{W}(h^{-1})}$ has the symbol $-\varepsilon(k)\sigma^y$, which commutes with that of $\mathbf{H}_I$. This, in turn, implies

$$\big[e^{i\mathbf{W}(h^{-1})}\mathbf{H}e^{-i\mathbf{W}(h^{-1})}, \mathbf{H}_I\big] = 0. \tag{162}$$

We can then identify the folded Hamiltonian with

$$\mathbf{H}_F(h^{-1}) = e^{i\mathbf{W}(h^{-1})}\mathbf{H}e^{-i\mathbf{W}(h^{-1})} - \frac{Jh}{2}\mathbf{H}_I, \tag{163}$$

that is to say, its symbol reads

$$\hat{h}_F(e^{ik}) = \Big[\frac{Jh}{2} - \varepsilon(k)\Big]\sigma^y = Jh\frac{1 - \sqrt{1 - 2h^{-1}\cos k + h^{-2}}}{2}\sigma^y. \tag{164}$$

The unitary transformation is instead given by

$$\mathbf{B}_1(h^{-1}) = h\mathbf{W}(h^{-1}) \tag{165}$$

and, from $\mathbf{B}_{z_t}(h^{-1}) = e^{ih\mathbf{H}_I t}\mathbf{B}_1(h^{-1})e^{-ih\mathbf{H}_I t}$, it follows that only $\mathbf{A}_1(\kappa)$ is different from zero, its symbol reading

$$\hat{a}_1(e^{ik}) = -\frac{i}{4}h\log\frac{1 - h^{-1}e^{ik}}{1 - h^{-1}e^{-ik}}\frac{\sigma^z - i\sigma^x}{2}. \tag{166}$$

## C    Energy and scattering matrix

The bare energy of quasiparticle excitations can be calculated by considering the one-particle sector, $N = 1$. In addition, if a coordinate Bethe Ansatz can be built from such quasiparticles, their scattering properties are completely determined by the solution in the two-particle sector, $N = 2$, which is supposed to completely characterise the scattering properties. The other sectors are then solved by a Bethe Ansatz, which is presented in full generality in Section 5.

### One-particle sector

**Even number of sites.**    In the one-particle sector there are only two possible configurations, $B_1^{(e)} = \{(0)\}_c \equiv (0)$ and $B_1^{(e)} = \{(1)\}_c \equiv (1)$. Specifically, the sector is spanned by states of the form

$$|p\rangle_{(b)} = \sum_{\ell'=1}^{\frac{L}{2}} c_{\ell'}^b(p)\boldsymbol{\sigma}_{2\ell'-b}^x |\phi\rangle, \qquad |\phi\rangle = |\downarrow, \ldots, \downarrow\rangle, \tag{167}$$

where $b = 0$ or $b = 1$. Demanding $\tilde{\mathbf{H}}_F^\eta |p\rangle_{\{(b)\}_c} = E_{\{(b)\}_c}(p) |p\rangle_{\{(b)\}_c}$ and imposing the boundary conditions, we obtain

$$2J\left[c_{\ell'-1}^b(p) + c_{\ell'+1}^b(p)\right] = E_{\{(b)\}_c}(p)c_{\ell'}^b(p), \qquad c_{\frac{L}{2}+\ell'}^b(p) = (-1)^\eta c_{\ell'}^b(p). \tag{168}$$

This is solved by plane waves

$$c_{\ell'}^b(p) = \sqrt{\frac{2}{L}}e^{i\ell'p}, \qquad e^{i\frac{L}{2}p} = (-1)^\eta, \tag{169}$$

with energy

$$E_{\{(b)\}_c}(p) = 4J\cos p. \tag{170}$$

As expected by one-site shift invariance, the excitation energy does not depend on the particular configuration. Observe that no eigenstate in this sector is mapped into an eigenstate of $\tilde{\mathbf{H}}_F$: since $L$ is even, the vacuum belongs to the sector $\boldsymbol{\Pi}^z = 1$, hence the single-particle excitations of $\tilde{\mathbf{H}}_F^\eta$ are in the sector $\boldsymbol{\Pi}^z = -1$, which is not associated with eigenstates of $\tilde{\mathbf{H}}_F$ – *cf.* Eq. (41).

**Odd number of sites.**    Normally, one would look for a solution of the form

$$|k\rangle = \frac{1}{\sqrt{L}}\sum_{\ell=1}^{L} e^{i\ell k}\boldsymbol{\sigma}_\ell^x |\phi\rangle, \qquad e^{iLk} = (-1)^\eta, \tag{171}$$

which has energy $E(k) = 4J\cos(2k)$, as inferred from the action of $\tilde{\mathbf{H}}_F^\eta$. It is however more convenient to enforce the description in terms of particles lying on macrosites. From that perspective, we must distinguish between even and odd sites

$$|k\rangle = \frac{1}{\sqrt{L}}\sum_{\ell'=1}^{\frac{L-1}{2}} e^{i\ell'(2k)}\boldsymbol{\sigma}_{2\ell'}^x |\phi\rangle + \frac{e^{-ik}}{\sqrt{L}}\sum_{\ell'=1}^{\frac{L+1}{2}} e^{i\ell'(2k)}\boldsymbol{\sigma}_{2\ell'-1}^x |\phi\rangle. \tag{172}$$

Concerning the particle configuration, we note instead that, up to cyclic permutations, there is a single possibility: $\{(0;1)\}_c \equiv \{(0;1),(1;0)\}$. It is now natural to identify the momentum

of the particles with $p = 2k$, so that also the energy keeps the same form as in the even-size case, *i.e.*,

$$E_{\{(0;1)\}_c}(p) = 4J\cos p. \tag{173}$$

Quantisation condition for $p$ is obtained by squaring the one for $k$, that is,

$$e^{iLp} = 1. \tag{174}$$

This results in some ambiguity in the relation between $k$ and $p$, which can be lifted by imposing the boundary condition $\boldsymbol{\sigma}^x_{\ell+L} = (-1)^\eta \boldsymbol{\sigma}^x_\ell$. Indeed, using it in the second sum of Eq. (172) and demanding the uniqueness of the state, we obtain

$$e^{-ik} = (-1)^\eta e^{i\frac{L-1}{2}p} = (-1)^\eta e^{-i\frac{L+1}{2}p}, \tag{175}$$

where the quantisation condition (174) was imposed in the second equality. Finally, we have

$$|k\rangle \equiv |p\rangle_{\{(0;1)\}_c} = \frac{1}{\sqrt{L}}\sum_{\ell'=1}^{\frac{L-1}{2}} e^{i\ell' p}\boldsymbol{\sigma}^x_{2\ell'}|\phi\rangle + (-1)^\eta \frac{e^{-i\frac{L+1}{2}p}}{\sqrt{L}}\sum_{\ell'=1}^{\frac{L+1}{2}} e^{i\ell' p}\boldsymbol{\sigma}^x_{2\ell'-1}|\phi\rangle. \tag{176}$$

Remarkably, only the eigenstates depend on $\eta$ (the momenta do not). Since, in this sector, one has $\boldsymbol{\Pi}^z = 1$, each eigenstate is mapped into one of $\tilde{\mathbf{H}}_F$ through Eq. (41).

### Two-particle sector

**Even number of sites.** For $N = 2$ there are three distinct configurations: $\{(0,0)\}_c$, $\{(1,1)\}_c$, and $\{(1,0)\}_c$. They correspond to two particles of type $\oplus$, two particles of type $\ominus$, and two particles of different species, respectively. In fact, configurations $\{(0,0)\}_c \equiv (0,0)$ and $\{(1,1)\}_c \equiv (1,1)$ are mapped into one another by a one-site shift and hence share the same properties. Indeed, they belong to the noninteracting sectors, that are mapped to the eigenstates of the XX Hamiltonian (49) under transformation $\oplus \mapsto \uparrow$, $\varnothing \mapsto \downarrow$ (or $\ominus \mapsto \uparrow$, $\varnothing \mapsto \downarrow$). Their eigenstates are simply given by

$$|p_1, p_2\rangle_{(b,b)} = \sum_{\substack{\ell',n'=1 \\ \ell'<n'}}^{\frac{L}{2}} \left[c^b_{\ell'}(p_1)c^b_{n'}(p_2) - c^b_{\ell'}(p_2)c^b_{n'}(p_1)\right]\boldsymbol{\sigma}^x_{2\ell'-b}\boldsymbol{\sigma}^x_{2n'-b}|\phi\rangle, \qquad e^{i\frac{L}{2}p_j} = (-1)^\eta, \tag{177}$$

where the relative minus sign is due to the fermionic nature of the scattering process. The excitation energy is the sum of the single-particle excitation energies

$$E_{\{(b,b)\}_c}(p_1, p_2) = 4J\cos p_1 + 4J\cos p_2. \tag{178}$$

The case of configuration $B_2^{(e)} = \{(0,1)\}_c \equiv \{(0,1),(1,0)\}$ is more interesting. This subspace is spanned by states of the form

$$|p_1, p_2\rangle_{\{(0,1)\}_c} = \sum_{\substack{\ell',n'=1 \\ \ell'<n'}}^{\frac{L}{2}} c^{0,1}_{\ell',n'}(p_1,p_2)\boldsymbol{\sigma}^x_{2\ell'}\boldsymbol{\sigma}^x_{2n'-1}|\phi\rangle + \sum_{\substack{\ell',n'=1 \\ \ell'\leq n'}}^{\frac{L}{2}} c^{1,0}_{\ell',n'}(p_1,p_2)\boldsymbol{\sigma}^x_{2\ell'-1}\boldsymbol{\sigma}^x_{2n'}|\phi\rangle. \tag{179}$$

Note that the sum in the second term allows for the particles $\ominus$ on the left-hand side and $\oplus$ on the right-hand side, to share the same macrosite (we denote it by $\underline{\ominus\oplus}$). The Ansatz that we impose upon the coefficients is now

$$c^{b_1,b_2}_{\ell',n'}(p_1,p_2) = Z^{b_1,b_2}_{p_1,p_2}\left[e^{i\ell' p_1+in' p_2} + S_{b_1,b_2}\binom{p_1,p_2}{p_2,p_1}e^{i\ell' p_2+in' p_1}\right], \tag{180}$$

where $S_{b_1,b_2}\begin{pmatrix}p_1,p_2\\p_2,p_1\end{pmatrix}$ denotes the amplitude of the scattering process that exchanges the momenta $p_1$ and $p_2$. Imposing

$$\tilde{\mathbf{H}}_F^\eta |p_1,p_2\rangle_{\{(0,1)\}_c} = E_{\{(0,1)\}_c}(p_1,p_2)|p_1,p_2\rangle_{\{(0,1)\}_c} \tag{181}$$

we obtain the explicit form of the scattering matrix

$$S_{b_1,b_2}\begin{pmatrix}p_1,p_2\\p_2,p_1\end{pmatrix} = -1 + b_1(1-b_2)\left(1 - e^{i[p_1-p_2]}\right), \tag{182}$$

which includes the noninteracting case $b_1 = b_2$, considered before. By virtue of the factorisation of the multi-particle scattering processes, encoded in the celebrated Yang-Baxter equation

$$S_{b_1,b_2}\begin{pmatrix}p_2,p_3\\p_3,p_2\end{pmatrix}S_{b_2,b_3}\begin{pmatrix}p_1,p_3\\p_3,p_1\end{pmatrix}S_{b_1,b_2}\begin{pmatrix}p_1,p_2\\p_2,p_1\end{pmatrix} = S_{b_2,b_3}\begin{pmatrix}p_1,p_2\\p_2,p_1\end{pmatrix}S_{b_1,b_2}\begin{pmatrix}p_1,p_3\\p_3,p_1\end{pmatrix}S_{b_2,b_3}\begin{pmatrix}p_2,p_3\\p_3,p_2\end{pmatrix}, \tag{183}$$

this scattering matrix describes also the eigenstates for $N > 2$. Enforcing the boundary conditions

$$c_{\ell',n'}^{b_1,b_2}(p_1,p_2) = (-1)^\eta c_{n',\ell'+\frac{L}{2}}^{b_2,b_1}(p_1,p_2), \tag{184}$$

for generic $\ell'$ and $n'$ and $(b_1,b_2) \in \{(0,1)\}_c$, we then obtain

$$\frac{Z_{p_1,p_2}^{0,1}}{Z_{p_1,p_2}^{1,0}} = -(-1)^\eta e^{-i\frac{L}{2}p_1}, \tag{185}$$

as well as the Bethe equations

$$e^{i\frac{L}{2}(p_1+p_2)} = 1, \qquad e^{i\left(\frac{L}{2}+1\right)(p_1-p_2)} = 1. \tag{186}$$

Consistently with the applicability of the coordinate Bethe Ansatz, the energy is still written as the sum of the single-particle energies, *i.e.*,

$$E_{\{(0,1)\}_c}(p_1,p_2) = 4J\cos p_1 + 4J\cos p_2. \tag{187}$$

**Odd number of sites.** For $N = 2$, there is a single configuration $B_2^{(o)} = \{(0,0;1,1)\}_c \equiv \{(0,0;1,1),(1,0;0,1),(1,1;0,0),(0,1;1,0)\}$. The corresponding subspace is spanned by states of the form

$$|p_1,p_2\rangle_{\{(0,0;1,1)\}_c} = \sum_{b_1,b_2\in\{0,1\}} \sum_{\ell'=1}^{\frac{L-1}{2}+b_1} \sum_{n'=\ell'+\lceil\frac{b_2-b_1+1}{2}\rceil}^{\frac{L-1}{2}+b_2} c_{\ell',n'}^{b_1,b_2}(p_1,p_2)\sigma_{2\ell'-b_1}^x \sigma_{2n'-b_2}^x |\phi\rangle. \tag{188}$$

The Ansatz for the coefficients is the same as in the case of an even number of sites:

$$c_{\ell',n'}^{b_1,b_2}(p_1,p_2) = Z_{p_1,p_2}^{b_1,b_2}\left[e^{i\ell'p_1+in'p_2} + S_{b_1,b_2}\begin{pmatrix}p_1,p_2\\p_2,p_1\end{pmatrix}e^{i\ell'p_2+in'p_1}\right]. \tag{189}$$

The boundary conditions now account for the transmutation of the particles and read

$$c_{\ell',n'}^{b_1,b_2}(p_1,p_2) = (-1)^\eta c_{n',\ell'+\frac{L+1}{2}-b_1}^{b_2,1-b_1}(p_1,p_2), \tag{190}$$

for generic $\ell'$ and $n'$. From them we obtain

$$\frac{Z_{p_1,p_2}^{b_1,b_2}}{Z_{p_1,p_2}^{b_2,1-b_1}} = (-1)^\eta \left[ -1 + b_2 b_1 \left( 1 - e^{i[p_1-p_2]} \right) \right] e^{i\left(\frac{L+1}{2}-b_1\right)p_1} \tag{191}$$

and the Bethe equations

$$e^{iL(p_1+p_2)} = 1, \qquad e^{i\frac{L+1}{2}(p_1-p_2)} = 1. \tag{192}$$

As in the even case, the energy is the sum of the single-particle energies

$$E_{\{(0,0:1,1)\}_c}(p_1,p_2) = 4J\cos p_1 + 4J\cos p_2. \tag{193}$$

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
