# Peer review of "The Folded Spin-1/2 XXZ Model: I. Diagonalisation, Jamming, and Ground State Properties"

_SciPost Physics_

## Round 1 · Referee Report · Anonymous (Referee 1) · 2020-10-17

Strengths

1-It is a clear paper that introduces a procedure to study models in a restricted context of large ones.
2-The paper is interesting and clear.

Report

In this paper the authors provides a prescription to produce effective
Hamiltonians (folded Hamiltonian) that acts on a smaller Hilbert space
("restricted space") as compared with the original one. This folded
Hamiltonian is obtained by appropriate strong-coupling limit.

The main application in the paper is an XXZ-like chain (Eq. 27), where
pairs of nearest neighbors spins (up and down) interchange positions
only if the pair has as neighbors two equal spins. They made an detailed
study of the models's charge conservations, and obtained, that the ground state
belongs to the constrained sector where two up spins are not allowed. Since
restricted to this sector the Hamiltonian (27) is equivalent to the
constrained XXZ Hamiltonian solved in Ref.[20], their solution recovers that
of this mentioned reference.

In fact, it seems that the several Hilbert space sectors associated to (27) can be mapped
to the Hilbert spaces of a constrained model that generalizes the one in Ref.[20].
In this generalization we have a mixture of of clusters of l up spins
("molecule of size l"). We have a set of n cluster of up spins (c_1,...,c_n),
and on each of these clusters we have a number of s_i=,1,2,...,l_i (size l_i)
up spins. This general model, in the sectors where all the particles have
the same size t (t=1,2,3,...) reproduces the result of [20], being a special sector
of the Hamiltonian (27), in the case where t=2. This general Hamiltonian was introduced in order
to describe the asymmetric diffusion of a mixture of particles with arbitrary size.
It was solved by the coordinate Bethe ansatz in Alcaraz and Bariev, Phys. Rev. E (33) (1999),
and also by a Matrix-Product ansatz in Alcaraz and Lazo, Braz. J. Phys. (33) (2003).
An more general case related to the constrained spin-1 XXZ chains was also solved
in Alcaraz and Bariev Braz. J. Phys. (30) 655 (2000).

I think the paper is interesting and contains interesting results, and
present a way to see quantum chains in restricted Hilbert spaces.
For this reason it should be published, but it would be interesting if the authors
made a contact with the above mentioned considerations.
  • validity: high
  • significance: high
  • originality: high
  • clarity: high
  • formatting: good
  • grammar: good

Author:  Lenart Zadnik  on 2020-11-17  [id 1046]

(in reply to Report 1 on 2020-10-17)
Category:
remark
answer to question

Dear Referee,

Thank you for the review of our paper and positive feedback. We would like to clarify few points that
might have not been stressed enough in the submitted manuscript.

1.) The projection of the folded Hamiltonian (27) onto the subspace with forbidden occupation of two
neighbouring sites is equivalent to the analogous projection of the XX Hamiltonian, defined in Ref. [20].
The corresponding projector indeed commutes with the folded Hamiltonian and defines a sector containing
the ground state of the folded Hamiltonian. The situation is however a bit different with regard to the projectors
onto subspaces of states, in which two spins that are less than $r >1$ sites apart cannot simultaneously point
downwards. Upon additional checks we have observed that the projections of the folded Hamiltonian and of
the XX model onto such sectors still coincide, but the projectors themselves no longer commute with the folded
Hamiltonian (27). Thus, the particles with higher radii discussed in Ref. [20] do not coincide with any configuration
of our folded Hamiltonian, i.e., they do not define the sectors of Hamiltonian (27) in the submitted manuscript.
We have incorporated this into the discussion in Section 3.1, around Eqs (32) and (33) - see the newest version of
the manuscript, accessible on the arXiv: https://arxiv.org/abs/2009.04995

2.) Related to the previous observation, there is a point that, arguably, we didn’t stress enough: the Bethe Ansatz
that we presented produces $2^L$ eigenstates that, in turn, span the entire Hilbert space (including the kernel of
the Hamiltonian, which, in our case, includes a set of jammed states). This goes beyond the standard Ising limit of the
XXZ Bethe Ansatz, discussed in Refs [18,19], or the Bethe Ansatz used in Ref. [20] to diagonalise the sector with
prohibited nearest-neighbour occupation. To the best of our knowledge, the structure that we pointed out was not
investigated before, and we think that it provides a general framework that is suitable, if not even necessary, for
investigations into quench dynamics in the strong coupling limit. Appropriate comments have been inserted in the
introduction to Section 5.

3.) We have recently realised that the more general folded XYZ model was solved by Bariev using the nested Bethe
Ansatz method – it now goes under the name of Bariev model. Bariev’s solution does not rely on the additional
symmetries present at the XXZ point. The main difference between his and our Ansatz is in the quantum numbers:
in our case only one set of quantum numbers is necessary for the characterisation of the eigenstates; we somehow
circumvent the nested structure. We have added some comments and references into the text: see the last paragraphs
after Eqs (41) and (93) in the newest version of the manuscript.

4.) Finally, we would like to remark that the point of discussing the large-anisotropy limit of the Heisenberg
model lies in the fact that it provides access to the dynamics on intermediate-time scales, one of the goals
being the study of pre-relaxation phenomena.

We hope that the new version of the manuscript is clearer.

The Authors

---

## Round 1 · Referee Report · Paul Fendley (Referee 2) · 2020-12-30

Report

The authors first generalize an old procedure from MacDonald et al for doing a systematic expansion of a Hamiltonian around a limit of a single large coupling, doing what is often called a Schrieffer-Wolf transformation. They then analyze the particular case of the XXZ chain expanded around large $\Delta$. They find the striking result that the resulting zeroth order Hamiltonian is much simpler than generic models solvable by the coordinate Bethe ansatz -- the momenta can be determined exactly, as the bare scattering matrices cancel in the Bethe equations.

I think this result is very interesting and should be published in SciPost. My main comment is that both parts of the paper are closely related to earlier results, and very possibly some of the work done previously could be of use in simplifying and/or extending the authors' analysis.

I take the two parts in turn. For the first part, the procedure of expanding Hamiltonians as described in section 2 of this paper was developed in great depth by Abanin, de Roeck, Huveneers and Ho (arXiv:1509.05386, published in CMP). ADHH prove (as in rigorous) that such an expansion exists up to corrections almost exponentially small in the large parameter ($1/\kappa\,$ in this paper). The authors don't discuss the validity of their expansion at all, and so should mention this. In the integrable XXZ case described in depth here, I would guess the procedure will work to all orders, but that is not guaranteed.

Also, thinking about the expansion the way ADHH do probably will lead to some useful intution. ADHH think of the large term in the Hamiltonian as an symmetry (approximate if the series does not converge). In the case that the authors study, this symmetry amounts to domain-wall number conservation. Also note that the expansion for the Ising chain is given explicitly earlier in my paper with Else, Kemp and Nayak (arXiv:1704.08703, published in PRX), where it is also explained how in essence that this expansion was how Onsager solved the Ising model.

While ADHH explicitly construct the unitary transformation order by order, their expression is unwieldy, and so while useful for their proof, it is not so useful in understanding the resulting physics. So the authors' explicit analysis of the XXZ chain here is still well worth doing. In fact, even though the zeroth order Hamiltonian was already solved by Bariev, their method exhibits a feature not seen here before -- the Bethe equations can be solved explicitly, giving in essence a free fermion spectrum, except with unusual degeneracies.

This brings me to the second observation of my report, concerning the zeroth order approximation to XXZ that the authors devote the bulk of their paper to. A Hamiltonian with essentially the same behavior was analysed in depth in a paper I wrote with Schoutens a while ago (arXiv:cond-mat/0612270, published in JSTAT). Very possibly the authors here could get some useful intuition from these results. Moreover, since the model Zadnik and Fagotti study is simpler (the only interaction is that fermions hop only if the intermediate site is empty), possibly something could be learned about the earlier model as well. In the old model, the degeneracies were interpreted as arising from Cooper pairs, and the remaining "free" quasiparticles were understood as exclusons in the sense defined by Haldane. I'd guess a similar nice interpretation is possible here, and may simplify the analysis, which gets a little techinical. In addition, comparing the enhanced symmetry algebras giving the degeneracies (there it is a generalization of supersymmetry) possibly will lead to some useful intution and/or progress.
  • validity: -
  • significance: -
  • originality: -
  • clarity: -
  • formatting: -
  • grammar: -

Author:  Lenart Zadnik  on 2021-02-10  [id 1223]

(in reply to Report 3 by Paul Fendley on 2020-12-30)

Dear Prof. Fendley,

thank you for your constructive comments and positive feedback. We also thank you for having brought Refs arXiv:1509.05386, arXiv:1704.08703, and arXiv:cond-mat/0612270 to our attention; they are now appropriately acknowledged. We indeed think that the enhanced symmetry algebras you refer to might be of significance in our considerations, especially in connection with the non-abelian symmetry structure of our model.

In the following we provide some additional clarification.

Firstly, we would like to stress that the model that we studied is genuinely interacting. The mapping into free fermions is possible only within very small sectors, where constraints on the total momentum and on the effective length of the system arise. The scattering matrices do not cancel exactly, but they are simple enough to resemble the Bethe equations in noninteracting systems. The appropriate comments have already been added to the third arXiv version of the manuscript (arXiv:2009.04995v3), which was unfortunately not linked on the Submissions webpage.

The goal of our paper is to provide a simple Bethe ansatz solution that is not restricted to a particular sector. As noted in the report, the model itself is a special point of the Bariev model, which was solved via a nested Bethe ansatz with two sets of quantum numbers, related to two species of particles. In this respect our solution (which describes all sectors) is somewhat simpler and more convenient for the thermodynamic Bethe ansatz description investigated in the second part of our work (arXiv:2011.01159v3).

Yours sincerely,

The Authors

---

## Editorial Decision

resubmitted